# LoTa-Bench: Benchmarking Language-oriented Task Planners for Embodied Agents

**Jae-Woo Choi**[1*], **Youngwoo Yoon**[1*], **Hyobin Ong**[1,2], **Jaehong Kim**[1], **Minsu Jang**[1,2]
[1] Electronics and Telecommunications Research Institute
[2] University of Science and Technology
{jwchoi0717,youngwoo,ohnghb,jhkim504,minsu}@etri.re.kr

## Abstract

Large language models (LLMs) have recently received considerable attention as alternative solutions for task planning. However, comparing the performance of language-oriented task planners becomes difficult, and there exists a dearth of detailed exploration regarding the effects of various factors such as pre-trained model selection and prompt construction. To address this, we propose a benchmark system for automatically quantifying performance of task planning for home-service embodied agents. Task planners are tested on two pairs of datasets and simulators: 1) ALFRED and AI2-THOR, 2) an extension of Watch-And-Help and VirtualHome. Using the proposed benchmark system, we perform extensive experiments with LLMs and prompts, and explore several enhancements of the baseline planner. We expect that the proposed benchmark tool would accelerate the development of language-oriented task planners.

## 1 Introduction

The ability of embodied agents to comprehend natural language instructions and perform the desired tasks has been a long-standing goal in the field of AI and robotics. When the agent has a sufficiently diverse skill set, decomposing high-level tasks into sequences of executable skills becomes particularly important. Conventional approaches have addressed this challenge through symbolic planning in predefined domains (Fikes & Nilsson, 1971; Garrett et al., 2020) or through learning-based task and motion planning (Silver et al., 2023; Shah et al., 2022; Li et al., 2022). Recently, large language models (LLMs) have emerged as a promising alternative. These models, pre-trained on extensive corpora, seem to have semantic knowledge about the world (Brown et al., 2020; Chowdhery et al., 2023; Thoppilan et al., 2022; Zhang et al., 2022). This knowledge can be effectively leveraged for high-level task planning through in-context learning without any additional training (Huang et al., 2022; Singh et al., 2023; Liang et al., 2023a; Ahn et al., 2023; Huang et al., 2023; Yao et al., 2023).

However, the evaluation frameworks for LLM-based task planning remain underdeveloped. Most existing studies rely on human evaluation, which is not only time-consuming but also expensive. These evaluations often occur in custom environments, which also makes them difficult to reproduce. Although some research (Huang et al., 2023; Liang et al., 2023a) has utilized simulators for automated evaluation, these efforts are typically confined to simple tabletop manipulation tasks. Furthermore, there is a noticeable absence of in-depth investigation into various influential factors, such as the type and size of pre-trained model, the number and select strategy of in-context examples, the capability for replanning based on natural language feedback, and the impact of fine-tuning.

To address the limitations, we introduce LoTa-Bench, a benchmark for language-oriented task planning for embodied agents. Our system aims to automatically quantify planning performance, enabling easier, fair, and reproducible comparison between systems. The framework consists of a baseline task planner, a dataset, and a simulator, as illustrated in Figure 1. The baseline task planner capitalizes on the in-context learning ability of LLMs. It constructs a prompt using a prefix, in-context examples (comprising pairs of natural language instructions and corresponding skill sequences to accomplish the instruction), and a user-provided natural language instruction. With this

---

*Equal contribution.

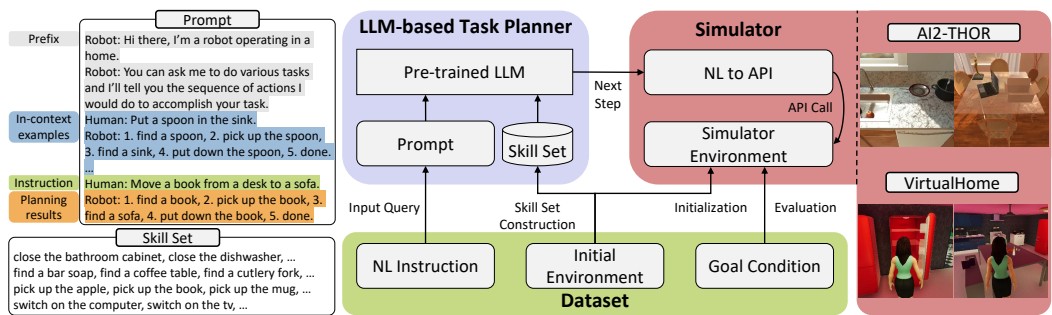

Figure 1: Overall benchmarking configuration for LLM-based task planners. NL stands for Natural Language. We used two setups: 1) ALFRED dataset with AI2-THOR simulator and 2) WAH-NL dataset with VirtualHome simulator. Exemplary prompt and skill set are presented on the left side.

prompt, the LLM calculates the probabilities of all executable skills to complete a task. The skill with the highest probability is selected and appended to the prompt for the next step in an autoregressive manner. In the proposed benchmark suite, we evaluate the planner on two dataset-simulator pairs: 1) ALFRED dataset (Shridhar et al., 2020) with AI2-THOR simulator (Kolve et al., 2017), and 2) our extension of Watch-And-Help (WAH) dataset (Puig et al., 2021), WAH-NL, paired with VirtualHome simulator (Puig et al., 2018). Each dataset sample furnishes the planner with both a natural language instruction and an environment context. The simulator executes the planned actions, and the performance of task planning is automatically assessed by comparing the final state of the simulator with predefined goal conditions.

In addition to the introduction of the benchmark suite, we provide extensive experimental results to further understand LLM-based task planning. Our baseline experiments explore the influence of various pre-trained models and their sizes. Given the critical impact of in-context examples on the performance of LLM-based task planners, we investigate the effect of number of examples and selection strategies. Additionally, we probe into other influential factors such as replanning according to the failure of a previous step and the effectiveness of model fine-tuning in the task planning domain.

Our contribution is fourfold: 1) first proposal of a benchmark suite that enables automatic evaluation of LLM-based task planners for home-service agents, 2) extensive experiments of a baseline task planner, 3) exploring possible extensions of the baseline planner and its validation with the proposed benchmark, and 4) public release of benchmark code and extended dataset (WAH-NL); they are available at `https://github.com/lbaa2022/LLMTaskPlanning`.

## 2 RELATED WORK

LLMs have demonstrated remarkable generalization capabilities through zero-shot or few-shot prompting (Brown et al., 2020), leading to a transformative impact on task planning. Traditional task planning methods predominantly focused on searching within predefined domains (Fikes & Nilsson, 1971; Garrett et al., 2020; Hoffmann, 2001) or learning trajectories (Silver et al., 2023; Shah et al., 2022; Li et al., 2022; Ichter et al., 2022; Nair & Finn, 2020; Eysenbach et al., 2019; Xu et al., 2019). However, thanks to LLMs, new language-oriented task planning methods have emerged. Huang et al. (2022) proposed a method where an LLM directly generates task plans via prompt engineering, with each generated step translated into an executable action using another language model. SayCan (Ahn et al., 2023) employed an LLM to score all predifined admissible actions, concurrently considering skill affordance through learned vision-based value functions. LLMs have also been adopted to generate executable robot codes using program-style inputs such as function descriptions (Liang et al., 2023a; Singh et al., 2023; Zelikman et al., 2023). Moreover, integrating context into LLM-based task planners has been shown to enhance planning efficacy (Huang et al., 2023; Yao et al., 2023; Chen et al., 2023; Lin et al., 2023; Wu et al., 2023).

Although numerous LLM-based task planners have emerged, standardized automatic performance evaluation methods are still scarce. Real robot experiments typically require time-intensive human

evaluations. In these setups, human raters determine the success or failure of planning (Ahn et al., 2023; Huang et al., 2023; Chen et al., 2023). When using simulators and datasets for evaluations, each task requires the goal condition and the natural language instruction. If a dataset lacks goal conditions, such as ActivityPrograms (Puig et al., 2018), human evaluation remains necessary (Huang et al., 2022; Zelikman et al., 2023). Similarly, datasets without natural language instruction, like Watch-And-Help (Puig et al., 2021), or simulators not offering high-level APIs, such as Behavior-1k (Li et al., 2023), cannot support language-oriented task planning. Only a few studies, akin to our benchmark suite, have incorporated automated evaluations. For instance, ReAct (Yao et al., 2023) utilized the ALFWorld (Shridhar et al., 2021) text-based game and the ALFRED dataset (Shridhar et al., 2020). ProgPrompt (Singh et al., 2023) engaged with the VirtualHome simulator and a customized dataset. Nevertheless, these assessments were conducted in restricted settings, hindering insights into comprehensive potential of LLM-based task planners.

## 3 BASELINE LLM-BASED TASK PLANNER

**Problem Statement.** In our proposed framework, a task planner receives a natural language instruction $i$ from the user, e.g., "bring an apple and a cupcake and put them on the coffee table." The planner also has access to a predefined skill set $S$, where each skill $s \in S$ represents an atomic action the agent can perform, such as "pick up the apple," "find a wine glass," or "open the fridge." We assume that these skills are coupled with corresponding language-conditioned low-level controllers (Jang et al., 2022; Brohan et al., 2023). The objective of the task planner is to select the skill $s_t$ at time step $t$ by maximizing the likelihood of completing the given instruction $i$ as follows:

$$s_t = \arg\max_{s \in S} p(s|i, s_0, \cdots, s_{t-1}), \tag{1}$$

where $s_0, \cdots, s_{t-1}$ are previously executed skills and $s_0 = \emptyset$. Exemplary step sequences for the instruction we mentioned above could be (1. find an apple, 2. pick up the apple, 3. find a coffee table, 4. put down the apple, 5. find a cupcake, 6. pick up the cupcake, 7. find a coffee table, 8. put down the apple, 9. done).

**Baseline Task Planner.** Our baseline task planner leverages the in-context learning capabilities of large language models (LLMs), resonating with recent research (Huang et al., 2022; Liang et al., 2023a; Ahn et al., 2023). To estimate the probability expressed in Equation 1, we construct a prompt $P$, which consists of a prefix, in-context examples, the instruction $i$, and a history of previously executed skills. For a skill $s$, described by $n_s$ subword tokens $s = (w_1^s, \cdots, w_{n_s}^s)$, the LLM computes the conditional probability as follows:

$$p(s|i, s_1, \cdots, s_{t-1}) = p_{\text{LLM}}(s|P) = \prod_{n=1}^{n_s} p_{\text{LLM}}(w_n^s|P, w_0^s, \cdots, w_{n-1}^s), \tag{2}$$

where $p_{\text{LLM}}$ is the pre-trained LLM and $w_0^s = \emptyset$. Instead of iterating every skill to find the best next skill to perform (Equation 1), which requires extensive computation, we employ a greedy search strategy, but with constraints on the next token selection to match with one of the skills using Guidance tool (Microsoft, 2023); see Appendix B for details. Once a skill is selected, it is appended to the prompt $P$, and the planner continues to use the updated prompt to select the next skill. This autoregressive process continues until either the terminal skill ("done") is selected or the skill sequence reaches a predefined maximum length.

## 4 BENCHMARK SETUP

To rigorously evaluate LLM-based task planners, we introduce a comprehensive evaluation framework, described in Figure 1. The framework integrates three key components: a task planner, a dataset, and a simulator. The baseline task planner elaborated in Section 3 is employed for comparative benchmarking. Then we offer two distinct dataset-simulator pairings: 1) the ALFRED dataset (Shridhar et al., 2020) built on the AI2-THOR simulator (Kolve et al., 2017), and 2) an extended version of the Watch-And-Help (WAH) dataset (Puig et al., 2021), named WAH-NL, incorporated with the VirtualHome simulator (Puig et al., 2018). Further details of the dataset and the simulator are described in the following subsections.

## 4.1 DATASET

Our benchmark employs two datasets, the ALFRED dataset (Shridhar et al., 2020) and our extension of the WAH dataset (Puig et al., 2021). Both datasets include sets of a natural language (NL) instruction, an initial environment state, and a goal condition for home environments. The NL instructions are user-provided and serve as inputs to the task planner of an autonomous agent. The initial environment states, containing object locations and states, are used for initialization of the simulator and for skill set construction in the task planning. The goal condition specifies the criteria for task completion. Planning performance is assessed by comparing the final state of the simulator with this goal condition after the execution of the last skill generated by the task planner.

**ALFRED.** It is a benchmark dataset for embodied AI agents that plan and execute primitive actions to perform household tasks, such as heating a mug cup, placing a salt shaker in a drawer, or putting vegetables in the fridge. This dataset was built on the AI2-THOR simulation environment. There are 7 task types of *Pick & Place, Stack & Place, Pick Two & Place, Clean & Place, Heat & Place, Cool & Place*, and *Examine in Light*. Among them, we excluded *Pick Two & Place* type in the evaluation because of missing capability of object instance recognition, which is required to accomplish this task type, in the LLM-based task planner.

**WAH-NL.** The original WAH dataset focuses on the challenges of AI agents assisting humans in household tasks. It consists of a *Watch* stage where agents observe human demonstrations to infer goals, and a *Help* stage where agents assist human in achieving those goals with minimal time steps. The dataset includes 5 task types of *Setup a dinner table*, *Put groceries*, *Prepare a meal*, *Wash dishes*, and *Read a book*. The goal condition of each task consists of multiple subgoals. For example, the goal condition of *Put groceries* task can be "INSIDE(cupcake, fridge): 2" and "INSIDE(apple, fridge): 1", where the numeric values indicate the number of objects.

Our extended version, WAH-NL, introduced two significant modifications on the *Help* stage of the original WAH dataset. First, we adjust the goal conditions, originally designed for human-AI collaborations, to suit autonomous agents. Additionally, we set the number of objects for all subgoals to 1, for similar reasons as with the ALFRED dataset (the lack of object instance recognition capability in our LLM-based task planner). Second, since the original dataset lacks NL instructions, which is must-needed element for language-oriented task planners, we collected them via the Prolific crowd-sourcing platform. The final dataset includes 416 instructions for the *train* set and 195 for the *test* set. More details about WAH-NL are described in Appendix E.

## 4.2 SIMULATOR

The simulator serves as an interactive environment that enacts the skills generated by the task planner. We first define a skill set for the task planner by combining available actions with optional parameters like target objects or receptacles. The skill set includes, for example, "find an apple," "turn on the faucet," "open the fridge", and "put down the pillow." Then, we simulated language-conditioned low-level controllers by mapping the skills to executable agent action APIs of the simulators.

Our approach primarily employs two types of skills: object navigation (ObjNav) and object interaction. We opted for ObjNav over low-level move actions (such as move forward and rotate) as ObjNav is well studied and would be considered as a unit capability of home-service agents (Duan et al., 2022). We assume that the agent is fully aware of object locations in the scene (practically, it can be accomplished by scene exploration with object map building). Object interaction skills are executable only when the interacting object is close to the agent. For example, "pick up the plate" is successful when the distance between the plate and agent is less than a predefined distance. We had some other assumptions for the object interaction skill. For the "put down" skill, the agent always put a holding object on a receptacle last visited. If there are multiple objects with the same class, the object closest to the agent is selected. The agent can hold one object in AI2-THOR and two objects in VirtualHome.

**AI2-THOR.** There are seven interaction actions–"pick up," "open," "close," "turn on," "turn off," "slice," and "put down"–and one navigation action "find." Among all combinations of action and optional parameters such as target objects or receptacles, we used 214 skills that were used at least once in the ground-truth trajectories in the *train* set.

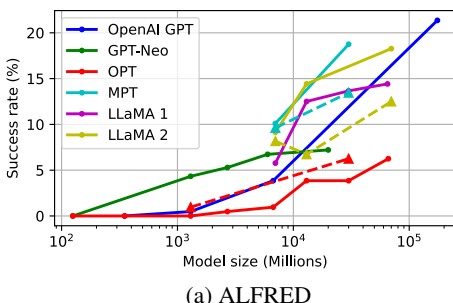 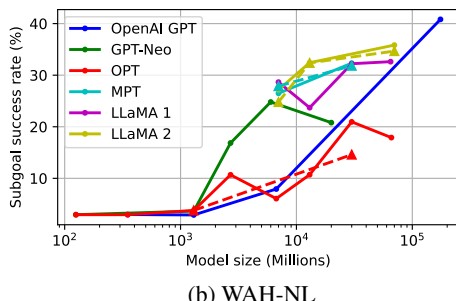

(a) ALFRED                                    (b) WAH-NL

Figure 2: Baseline results on (a) ALFRED and (b) WAH-NL. We report task success rates (%) on the ALFRED dataset and average subgoal success rate (%) on the WAH-NL datset for language models in different model classes and sizes (number of parameters). Base language models are represented as solid lines. Fine-tuned models (by either instruction or chat data) were shown in a dashed line with a triangle maker.

**VirtualHome.** This simulator supports five interaction actions – "pick up," "open," "close," "switch on," and "put down" – and one navigation action "find." In VirtualHome, due to the extensive total number of skills, we constructed a skill set by considering all possible combinations of actions and optional parameters for each environment. On average, we utilized 351.89 skills.

## 5    BASE EXPERIMENTS

We conducted experiments to measure the performance of the baseline LLM-based task planners by using the proposed benchmark. We tested various settings including LLMs in different model classes and sizes and the impact of the number of in-context examples.

### 5.1    EVALUATION PROTOCOL

**Test Setup.**  We used the ALFRED and WAH-NL datasets, as introduced in Section 4.1.  The ALFRED dataset consists of three sets: *train*, *valid-seen*, and *valid-unseen*. The *valid-seen* was used to evaluate planning performance; the *train* set was only used to take examples to construct prompts. We used a small set of *valid-seen*, which has 208 samples (30% of the *valid-seen* set), to accelerate the evaluation of various configurations (see Appendix C for the results with the full set).  The WAH-NL dataset comprises a *train* set and a *test* set with 250 and 100 samples, respectively. The *train* set was used to construct in-context examples, and the *test* set was used to evaluate planning performance.

The prompt comprises a prefix, describing the role of a home-service embodied agents, and a number of in-context examples, which adhere to the template defined in SayCan (Ahn et al., 2023).  Each example is a pair of an input query (task instructions) and the corresponding output step sequences to accomplish the task. See Appendix F for a complete prompt sample.  The in-context examples were randomly selected from the *train* sets, and we tried to use the same number of examples per task type. The default setup is to include six examples in ALFRED and five examples in WAH-NL (one example per task type).

**Evaluation Metrics.** We measured planning performance using the task success rates for ALFRED. Task success was determined based on whether the final state after executing the step sequence generated by the task planner satisfies the expected goal condition.  For WAH-NL, we measured the average subgoal success rate.  Each WAH-NL task has multiple subgoals.  We calculate the individual subgoal success rate as the ratio of successfully completed subgoals to the total number of subgoals for each task. The average subgoal success rate across a *test* set is reported.

### 5.2    BENCHMARK RESULTS OF BASELINE TASK PLANNER

We evaluated the planning performance of the baseline planner described in Section 3.  Figure 2 shows the results on ALFRED and WAH-NL for different pre-trained LLMs:  GPT (Brown

et al., 2020), GPT-Neo series (Black et al., 2021; Wang & Komatsuzaki, 2021; Black et al., 2022), OPT (Zhang et al., 2022), MPT (Team, 2023b;a), LLaMA 1 (Touvron et al., 2023a), and LLaMA 2 (Touvron et al., 2023b) (see Appendix A for the list). A few fine-tuned models on instructions or chat data were also tested.

Overall, task success rates increased with the size of the language model, but this was not always the case. For example, GPT-J 6B model performed better than GPT-NeoX 20B and OPT 2.7B was better than OPT 6.7B in the WAH-NL experiment. Such results, that a smaller model performs better than a larger model, were also observed in HELM evaluation (Liang et al., 2023b), especially in reasoning tasks. GPT-3 (text-davinci-003) showed the best success rate of 21.36% on ALFRED and the best subgoal success rate of 40.82% on WAH-NL. LLaMA 2 and MPT performed well considering their model sizes. Instruction- and chat-tuned models (dashed lines in Figure 2) did not perform better than their base models. We also tested on GPT-4, the state-of-the-art LLM. As OpenAI provides only chat-style APIs for GPT-4 unlike other base models such as GPT-3, we were unable to directly compare GPT-4 in the same configuration. We modified experimental configurations and assessed GPT-4's performance. GPT-4 performed well in ALFRED, showing a 40.38% success rate, a 19% improvement over GPT-3. However, in WAH-NL, GPT-4 showed a lower success rate of 34.17% compared to GPT-3. More details in Appendix D.

We conducted a further analysis of the task types. For ALFRED, we found that the small model such as GPT-J 6B succeeded only for simple *Pick & Place* tasks and failed in complex tasks such as heating and cooling tasks, which require longer steps than simple tasks. The largest model, GPT-3 175B, succeeded similarly in both simple and complex tasks (20-30% success rates), except for the task type *Stack & Place* where the agent needs to stack multiple objects in order. Additional results are shown in Table 7 in Appendix. For WAH-NL, all task types have a similar level of complexity, generally requiring the finding and placement of multiple objects. This resulted in a more balanced performance across task types when compared to ALFRED. Using our GPT-3 175B model, *Put Fridge* tasks yielded the highest average subgoal success rate at 54.50%, while *Prepare Snack* tasks registered the lowest average subgoal success rate of 25.00%. See Figure 4 for the success samples (more results in Appendix H).

We also examined the detailed reasons for the failure cases of the ALFRED results using GPT-3 model, which showed the highest performance. Out of 162 failure cases, the reasons were categorized into six classes: 1) Action planning failures, such as performing 'Pick' instead of 'Slice' when a tomato needs to be sliced. 2) Object selection failures, like grabbing a pan instead of a pot. 3) Absence of visual grounding, for instance, trying to grab an object inside a closed drawer, 4) Lack of physical understanding, such as failing to put down

| Failure category | # Failures |
|---|---|
| *Action planning failure* | 46 (28.4%) |
| *Object selection failure* | 51 (31.5%) |
| *Absence of visual grounding* | 21 (13.0%) |
| *Lack of physical understanding* | 15 (9.3%) |
| *Misunderstanding inst.* | 10 (6.2%) |
| *Ambiguous/incorrect inst.* | 19 (11.7%) |

Table 1: The number of failure cases of the ALFRED results using GPT-3.

an object on the already occupied table. 5) Misunderstanding user instructions, failing to distinguish between a desk lamp and a floor lamp when the user specified 'Lamp'. 6) Ambiguous or incorrect user instructions, like confusing 'Glass' for 'Cup' in an instruction. The results are presented in Table 1. Most failures (about 60%) stemmed from high-level planning (classes 1 and 2). Challenges in visual and physical grounding (classes 3 and 4) highlight the importance of integrating context in planning, as discussed in Section 7. Furthermore, the role of clarity in user instructions (classes 5 and 6) opens up a new research direction for interactive clarification of an ambiguous tasks.

We have investigated the impact of the number of examples in prompt with LLaMA 2 13B model that supports a longer context length of 4096. The success rate mostly increased when there are 0 to 30 examples on ALFRED and 0 to 15 examples on WAH-NL (see Figure 3). It was not able to test more than 15 examples on WAH-NL because of the maximum token limitation. Note that the pool of examples was fixed, which means that, for example, 6 and 12 examples share the same 6 examples. An additional experiment was performed to see whether different sets of examples matter for the same number of examples.

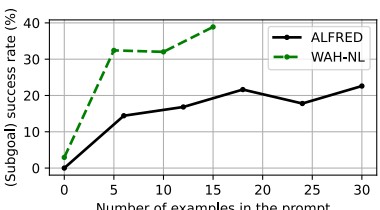

Figure 3: (Subgoal) success rates for the different number of examples for in-context learning.

(a) ALFRED: success example for *Cool & Place* task.

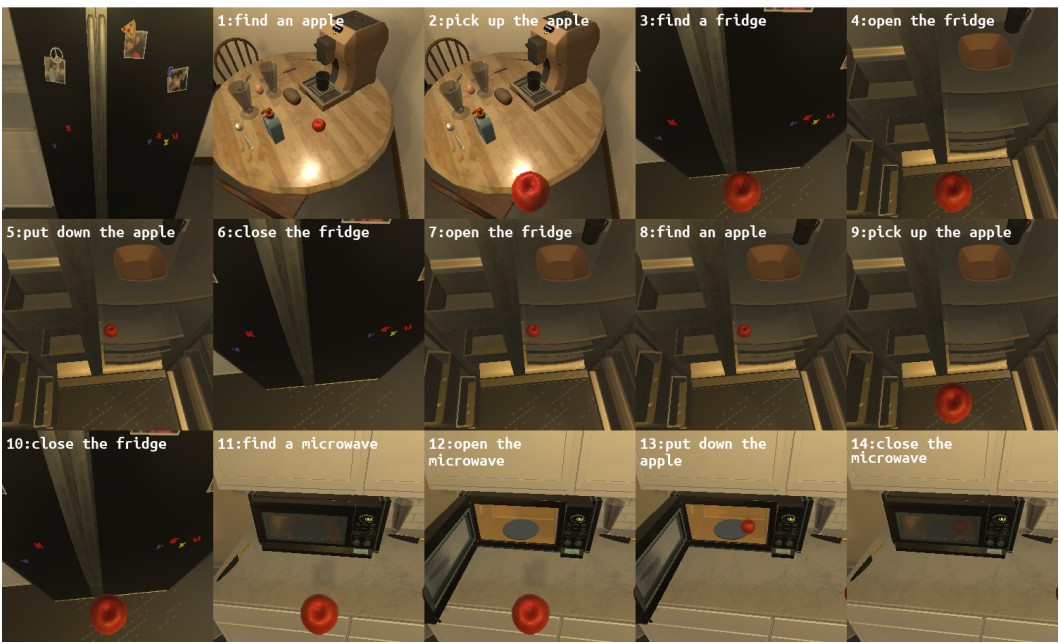

(b) WAH-NL: success example for *Wash dishes* task.

Figure 4: Planning results. Success cases of (a) the ALFRED task and (b) the WAH-NL task when GPT-3 175B model was used. The input instructions, inferred steps, and scene images after each step execution are presented in each figure. The scene images show agent's point of view on AL-FRED and third person's point of view on WAH-NL. Additional results, including failure cases, are provided in Appendix H.

We selected different sets of 12 examples that were randomly drawn from the example pool of the training set. The performance on ALFRED varied from a minimum of 9.62 to a maximum of 17.79 (with an average of 13.61 and a standard deviation of 3.22) for the LLaMA 2 13B model.

Additionally, we tested on both full sets of *valid-seen* and *valid-unseen* splits of ALFRED using LLaMA 2 13B model. *Valid-unseen* set contains scenes not present in the train set. The success rate on *valid-unseen* was similar (17.70%) to the one on *valid-seen* (17.82%) as two sets are different in visual scenes, not in task types.

## 6 VALIDATING EXTENSIONS OF THE TASK PLANNER

The primary merit of the proposed benchmark is that it allows faster and easier validation of new task planners. To demonstrate this, we explore some extensions (or improvements) of the base planner and validate them.

### 6.1 IN-CONTEXT EXAMPLE SELECTION

We explored three strategies for selecting in-context examples from the *train* set containing both input queries and their associated planning examples. The first strategy, named *Random Sampling*, is the same as that used in the baseline task planner. It involves random sampling of $N$ examples for each unique task type, leading to a collection of $N \times M$ examples across different task types, where $M$ is the number of distinct task types. In the second strategy, termed *Task-Specific Sampling*, we select examples from the subset of the *train* set that share the same task type as the input instruction (with the assumption that the task type is known). Lastly, *Semantic Similarity Sampling* employs Sentence BERT (Reimers & Gurevych, 2019), to compute the similarity scores between the input instruction and all instructions in the *train* set. The examples with the highest similarity scores are selected as in-context examples. This strategy not only aims to select the most relevant planning examples, but also offers potential utility in real-world scenarios where task types may not be explicitly provided.

To measure the effectiveness of these in-context example selection methods, we conducted experiments on WAH-NL using LLaMA 1 models. For all strategies, we set the number of in-context examples to five. The results are summarized in Figure 5. Across all model sizes, *Semantic Similarity Sampling* showed superior performance, followed by *Task-Specific Sampling*, and lastly *Random Sampling*. Importantly, *Semantic Similarity Sampling* led to significant performance gains; notably, the LLaMA 1 65B model achieved a subgoal success rate of 43.25%, surpassing the best performance of 40.82% achieved by our baseline using GPT-3 175B. These experiments confirm that the in-context example selecting strategy has a significant impact on the performance of LLM-based task planners. We found similar results with other LLM models like GPT-Neo and LLaMA 2.

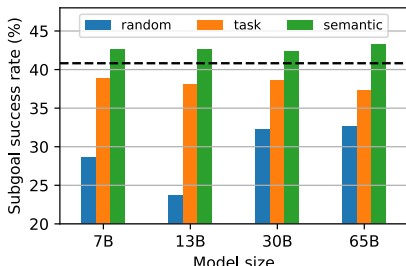

Figure 5: Subgoal success rate for different in-context example selection strategies. The dashed line represents the best performance of our baseline planner using GPT-3 175B.

### 6.2 FEEDBACK AND REPLANNING

The baseline planner selects the skill for the next step independent of the success or failure of previous action. However, adjusting the plan in response to the failure of the action is necessary for task planning in the wild. Using the ALFRED and AI2-THOR configuration, we investigated whether our LLM-based task planner can reflect feedback from action failure and replan appropriately. We added natural-language (NL) feedback at the end of each inferred step only when the step is failed. NL feedback is constructed based on error messages from the AI2THOR simulator and environment states. For instance, when the pick up action failed because the target object is inside of a container, *"(this action failed: [Object] is not visible because it is in [Container])"* is appended after the step failed in the prompt, and the next step is inferred afterward (see more information of feedback message in Appendix F). We assumed that the agent is fully aware of the location of objects. Exam-

ples demonstrating feedback and replanning were added to the prompt for in-context learning. We manually crafted 3 replanning examples and added them after the baseline prompt of 18 examples. The task instructions for the replanning examples were selected from the existing 18 examples to minimize the addition of task information (Listing 3 in Appendix F shows the additional in-context examples for replanning). LLaMA 2 model was used in this experiment.

Table 2 shows that replanning is helpful to improve the overall planning perofrmance when a large model, LLaMA 2 70B, is used. A smller model, LLaMA 2 13B, did not show improvements; this might be due to limited capability of the smaller model to understand complex concepts of task planning and replanning only with a few examples. The qualitative results showing how the planner succeeded by replanning steps are shown in Figure 12 in Appendix F.

|  | Success Rate(%) | |
| --- | --- | --- |
|  | LLaMA 2 13B | LLaMA 2 70B |
| Baseline | **21.15** (44/208) | 21.63 (45/208) |
| Replanning | 17.79 (37/208) | **24.04** (50/208) |

Table 2: Results with and without replanning. Success rates on ALFRED are reported.

### 6.3 FINE-TUNING ON TRAIN SET

We conducted experiments to investigate the potential to improve planner performance through fine-tuning. We fine-tuned LLaMA 1 models using LoRA (Hu et al., 2021) on the ALFRED *train* set and evaluated their performance in the same ALFRED domain both for *valid-seen* and *valid-unseen* tasks. For the evaluation of the fine-tuned planners, we used zero in-context examples as the planner had been explicitly trained. As depicted in Figure 6, fine-tuning significantly improved performance, especially for larger models, e.g. LLaMA 1 30B, which showed a performance jump from 13.66% to 60.08%. However, the planner, fine-tuned on ALFRED, performed significantly worse than the base-

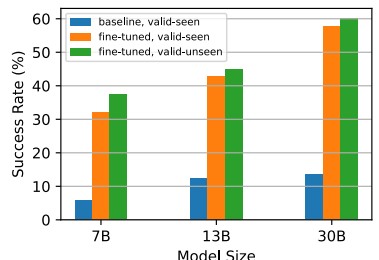

Figure 6: Success rates of fine-tuned planners on ALFRED.

line planners in the WAH-NL domain (32.22% → 10.38%), suggesting that procedural knowledge trained in a task domain does not transfer well to different task domains. See Appendix G for fine-tuning details and results. Appendix G also presents WAH-NL fine-tuning results. Notably, the limited number of WAH-NL *train* set led to marginal performance improvements. Specifically, the LLaMA 1 13B and 30B models showed increases of 5.77 and 1.56 percent points, respectively.

## 7 CONCLUSION AND LIMITATION

The impressive generalizability and performance of the large language model (LLM) has facilitated its versatile deployment across multiple domains. Task planning for embodied agents is one such application, and after the pioneering work proposed to use LLM for this application by Ahn et al. (2023) and Huang et al. (2022), we believe that diverse research efforts will continue in this direction. However, there was few way to automatically evaluate planners. In this paper, we proposed a quantitative evaluation benchmark for LLM-based task planning research to catalyze the rapid advancement of this field. The results of the base experiments and extensions of in-context example selection, replanning, and fine-tuning would be helpful to future studies. We hope that our proposed benchmark framework will serve as a starting point for the development of various extended models for language-oriented task planners.

The present work has the following limitations. First, we decoupled high-level plans and low-level actions to focus on high-level planning. An extension is needed to support an end-to-end system that considers both high-level planning and low-level actions. Visual understanding (egocentric views) is also necessary for low-level actions. Second, as is common in many simulator-based studies, there exists a domain gap between simulation and the real world. For example, ALFRED poses an unrealistic assumption that an object is always cleaned when it was once put into water in a sink basin. Furthermore, although AI2THOR and VirtualHome support multiple scenes and objects, they still lack diversity to reflect real-world environments. Future work will be to extend the proposed benchmark to support a wider range of systems and to reduce domain gaps.

REPRODUCIBILITY STATEMENT

We provide the source code of the benchmark suite and the configurations to reproduce the experimental results. WAH-NL dataset that we extended from the original WAH dataset to have task instructions is also available. See `https://github.com/lbaa2022/LLMTaskPlanning` for the source code and WAH-NL dataset.

ACKNOWLEDGMENTS

This work was supported by Institute of Information & communications Technology Planning & Evaluation (IITP) grant funded by the Korea government (MSIT) (No. 2022-0-00951, Development of Uncertainty-Aware Agents Learning by Asking Questions).

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

# A LANGUAGE MODELS

Table 3 lists the language models used in the experiments.

Table 3: List of language models used in the experiments. Model names are either from OpenAI API or HuggingFace model hub.

| Class | Model name | Model size | Remark |
|---|---|---|---|
| OpenAI GPT | ada | 350M | |
| | babbage | 1.3B | |
| | curie | 6.7B | |
| | text-davinci-003 | 175B | |
| GPT Neo | EleutherAI/gpt-neo-125m | 125M | |
| | EleutherAI/gpt-neo-1.3B | 1.3B | |
| | EleutherAI/gpt-neo-2.7B | 2.7B | |
| | EleutherAI/gpt-j-6b | 6B | |
| | EleutherAI/gpt-neox-20b | 20B | |
| OPT | facebook/opt-125m | 125M | |
| | facebook/opt-1.3b | 1.3B | |
| | facebook/opt-2.7b | 2.7B | |
| | facebook/opt-6.7b | 6.7B | |
| | facebook/opt-13b | 13B | |
| | facebook/opt-30b | 30B | |
| | facebook/opt-66b | 66B | |
| | facebook/opt-iml-max-1.3b | 1.3B | Instruction-tuned |
| | facebook/opt-iml-max-30b | 30B | Instruction-tuned |
| MPT | mosaicml/mpt-7b | 7B | |
| | mosaicml/mpt-30b | 30B | |
| | mosaicml/mpt-7b-instruct | 7B | Instruction-tuned |
| | mosaicml/mpt-30b-instruct | 30B | Instruction-tuned |
| LLaMA 1 | huggyllama/llama-7b | 7B | |
| | huggyllama/llama-13b | 13B | |
| | huggyllama/llama-30b | 30B | |
| | huggyllama/llama-65b | 65B | |
| LLaMA 2 | meta-llama/Llama-2-7b-hf | 7B | |
| | meta-llama/Llama-2-13b-hf | 13B | |
| | meta-llama/Llama-2-70b-hf | 70B | |
| | meta-llama/Llama-2-7b-chat-hf | 7B | Chat-tuned |
| | meta-llama/Llama-2-13b-chat-hf | 13B | Chat-tuned |
| | meta-llama/Llama-2-70b-chat-hf | 70B | Chat-tuned |

# B IMPLEMENTATION DETAILS OF THE BASELINE PLANNER

The implementation of SayCan's scoring LM (Ahn et al., 2023) calculates each skill probability by summing token log probabilities for each skill, so the calculation time increases proportionally to the number of skills. Unlike SayCan's implementation, we employed Guidance (Microsoft, 2023) that supports the selection of one skill among candidates from a single generation pass by penalizing tokens that did not appear in any skill candidates, which considerably reduces the total time needed for the experiments. Guidance also avoids tokenization artifacts, by so-called "token healing." We compared two implementation methods by their success rates and processing time. When we use GPT-Neo 6B on WAH-NL, the SayCan's implementation yielded a subgoal success rate of 21.65% and took 1649.2 minutes. Our implementation using Guidance achieved a success rate of 24.80% and took 87.4 minutes which is about 19x speedup.

We used HuggingFace's Transformers library (Wolf et al., 2020) and OpenAI's GPT API (OpenAI, 2023) for the experiments, along with Guidance library (Microsoft, 2023). Most of the models were run on a single NVIDIA A100 80GB GPU, while we used two A100 GPUs and three RTX 6000 GPUs for inference of larger models, such as OPT 66B and LLaMA 2 70B, with model parallelism.

## C EVALUATION WITH THE FULL SET OF ALFRED

The 30% subset we used in the main experiment with ALFRED might have some biases in the selection of subsets, although we randomly selected the samples. Thus, we report the results with the full set of ALFRED for validation. Note that GPT models were not included in this experiment as the lagacy GPT base models were deprecated.

Figure 7 shows the results with the full set and subset, and we found that the general trends are similar between two sets.

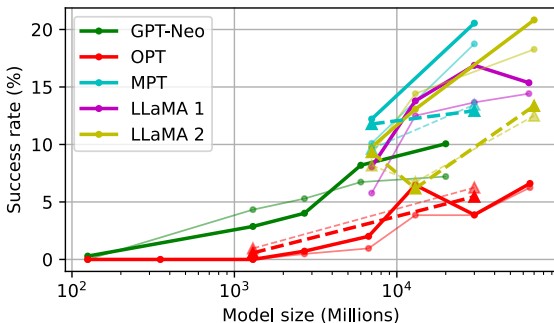

Figure 7: Baseline results on ALFRED with the full set and subset. We report task success rates (%) for language models in different model classes and sizes (number of parameters). The bold lines are for the full set. The thin lines represent the subset results. Fine-tuned models (by either instruction or chat data) were shown in a dashed line with a triangle maker.

## D GPT-4 EXPERIMENT DETAILS

GPT-4 API has three agent roles of "system," "user," and "assistant." We modified the original prompt (Listing 1) to fit these roles as follows. It worked like a prompt that gives the context of a problem and the role of the robot agent. The number of examples provided for in-context learning was the same. Examples, robot skills, and task query in square brackets [] were replaced with actual texts according to the test sample and environment.

<System Role> You are a robot operating in a home. A human user can ask you to do various tasks and you are supposed to tell the sequence of actions you would do to accomplish your task.
<User Role> Examples of human instructions and possible your (robot) answers: [The same in-context-learning examples are provided]
Now please answer the sequence of actions for the input instruction. You should use one of actions of this list: [List of robot skills provided] List the actions with comma separator.
Input user instruction: [Task query]

An assistant (robot) agent generates action sequences for the given user instruction. Due to the unavailability of Guidance's token selection mechanism Microsoft (2023) for GPT-4, we provided an admissible action list in the prompt to circumvent the issue.

GPT-4 generated the answer for the given user instruction, and we extracted each step from the answer text. There was no additional post-processing tailored to GPT-4 (steps not in the provided list of the skills would not be executed correctly).

We tested GPT-4 and GPT-3.5-Turbo (see the table below for the results). GPT-3.5-Turbo showed lower success rates for both ALFRED and WAH-NL compared to GPT-3. GPT-4 performed well in

ALFRED, showing a success rate of 40.38% and an improvement of 19% over GPT-3. However, in WAH-NL, GPT-4 showed a lower success rate of 34.17% compared to GPT-3.

Table 4: Success rates of GPT-4 and GPT-3.5 for ALFRED and WAH-NL datasets.

| Model | ALFRED | WAH-NL |
|---|---|---|
| GPT-3 (text-davinci-003) | 21.36% | 40.82% |
| GPT-4 | 40.38% | 34.17% |
| GPT-3.5-Turbo | 10.58% | 28.82% |

Despite the limitations of the current setup, GPT-4's performance notably surpassed that of GPT-3 in ALFRED. This observation leads us to anticipate that the incorporation of Guidance's token selection mechanism Microsoft (2023) for GPT-4 could yield more impressive planning capabilities.

# E  MORE DETAILS ON WAH-NL

WAH-NL comprises 5 task types: *Setup a dinner table*, *Put groceries*, *Prepare a meal*, *Wash dishes*, and *Prepare snacks*. These tasks are adapted from the orignal WAH dataset. In WAH, the *Wash dishes* task was deemed successful when dishes were placed in the dishwasher. We enhanced this by adding a subgoal, "SWITCHON(dishwasher): 1," ensuring the dishwasher is activated. The *Read a book* task in WAH, which implies a human reading a book while enjoying snacks, has been transformed into *Prepare snacks* task. This transformation involved the removal of two inappropriate subgoals, "HOLD(human, book): 1" and "SIT(human, sofa): 1." Moreover, the number of objects for all subgoals was set to 1 since our LLM-based task planner does not consider object instance recognition. Table 5 shows samples for these 5 task types.

We collected NL instructions for both the *train* and *test* sets, comprising 250 and 100 samples respectively, using the Prolific crowdsourcing platform. Crowdworkers were provided with 10 different goal conditions and asked to generate the corresponding user instructions. Initially, we collected 1,000 instructions, 712 for *train* set and 288 for *test* set. We eliminated erroneous samples that misinterpreted the goal conditions, resulting the final dataset of 416 instructions for the *train* set and 195 for the *test* set. Every sample in both sets has at least one NL instruction. Examples of the instructions collected are presented in Table 5.

Table 5: Task types and samples for each type in the WAH-NL datset.

| Task Type | Goal Condition | Instruction |
|---|---|---|
| *Setup a dinner table* | ON(plate, kitchen table): 1, ON(water glass, kitchen table): 1, ON(wine glass, kitchen table): 1, ON(cutlery fork, kitchen table): 1 | put the following on the kitchen table - 1 cutlery fork, 1 wine glass, 1 water glass and one plate |
| *Put groceries* | INSIDE(cupcake, fridge): 1, INSIDE(pancake, fridge): 1, INSIDE(pound cake, fridge): 1, INSIDE(apple, fridge): 1 | Please get the apple, the pancake, the pound cake and the cupcake and put them all in the fridge. |
| *Prepare a meal* | ON(pancake, kitchen table): 1, ON(pudding, kitchen table): 1 | Robot, please put the pancake and pudding on the kitchen table. |
| *Wash dishes* | INSIDE(plate, dishwasher): 1, INSIDE(wine glass, dishwasher): 1, SWITCHON(dishwasher):1 | Place one wine glass and one plate in the dishwasher and turn it on. |
| *Prepare snacks* | ON(juice, coffee table): 1, ON(apple, coffee table): 1 | Put one cupcake and one apple on the coffee table |

# F   PROMPT

We list the prompts used in the ALFRED and WAH-NL experiments. We also share the prompt of the replanning experiment, where replanning examples are added.

Listing 1: Prompt for ALFRED experiments when 18 in-context examples are provided (3 examples per task type). Subsets of the examples below are used when we use fewer examples, like 6 or 12 examples, in the base experiments.

Robot: Hi there, I'm a robot operating in a home.
Robot: You can ask me to do various tasks and I'll tell you the sequence of actions I would do to accomplish
     ↪ your task.
Human: Put a spoon in the sink.
Robot: 1. find a ladle, 2. pick up the ladle, 3. find a sink, 4. put down the ladle, 5. done.
Human: Move vase from the entertainment center to the coffee table.
Robot: 1. find a vase, 2. pick up the vase, 3. find a coffee table, 4. put down the vase, 5. done.
Human: Move the newspaper in the living room to the couch.
Robot: 1. find a newspaper, 2. pick up the newspaper, 3. find a sofa, 4. put down the newspaper, 5. done.
Human: Pick up a tennis racket and turn on a lamp.
Robot: 1. find a tennis racket, 2. pick up the tennis racket, 3. find a desk lamp, 4. turn on the desk lamp, 5.
     ↪ done.
Human: Pick up a pillow and turn on a lamp.
Robot: 1. find a pillow, 2. pick up the pillow, 3. find a desk lamp, 4. turn on the desk lamp, 5. done.
Human: Examine a credit card using the light from a floor lamp.
Robot: 1. find a credit card, 2. pick up the credit card, 3. find a floor lamp, 4. turn on the floor lamp, 5. done.
Human: Put a pan with a spoon in it on the table.
Robot: 1. find a spoon, 2. pick up the spoon, 3. find a pot, 4. put down the spoon, 5. pick up the pot, 6. find a
     ↪ dining table, 7. put down the pot, 8. done.
Human: Put a box with remote in it on the green couch.
Robot: 1. find a remote control, 2. pick up the remote control, 3. find a box, 4. put down the remote control, 5.
     ↪  pick up the box, 6. find a sofa, 7. put down the box, 8. done.
Human: Place a box with a cell phone on a chair.
Robot: 1. find a cell phone, 2. pick up the cell phone, 3. find a box, 4. put down the cell phone, 5. pick up the
     ↪ box, 6. find an arm chair, 7. put down the box, 8. done.
Human: Place a cold potato on the table.
Robot: 1. find a potato, 2. pick up the potato, 3. find a fridge, 4. open the fridge, 5. put down the potato, 6.
     ↪ close the fridge, 7. open the fridge, 8. find a potato, 9. pick up the potato, 10. close the fridge, 11.
     ↪ find a counter top, 12. put down the potato, 13. done.
Human: Put the chilled bowl in the microwave.
Robot: 1. find a cabinet, 2. open the cabinet, 3. find a bowl, 4. pick up the bowl, 5. close the cabinet, 6. find a
     ↪ fridge, 7. open the fridge, 8. put down the bowl, 9. close the fridge, 10. open the fridge, 11. find a
     ↪ bowl, 12. pick up the bowl, 13. close the fridge, 14. find a microwave, 15. open the microwave, 16.
     ↪ put down the bowl, 17. close the microwave, 18. done.
Human: Put the chilled sliced cabbage in the trash bin.
Robot: 1. find a knife, 2. pick up the knife, 3. find a lettuce, 4. slice the lettuce, 5. find a fridge, 6. open the
     ↪ fridge, 7. put down the knife, 8. close the fridge, 9. find a lettuce, 10. pick up the lettuce, 11. find a
     ↪ fridge, 12. open the fridge, 13. put down the lettuce, 14. close the fridge, 15. open the fridge, 16. find
     ↪  a lettuce, 17. pick up the lettuce, 18. close the fridge, 19. find a garbage can, 20. put down the
     ↪ lettuce, 21. done.
Human: Place a heated plate on the round table.
Robot: 1. find a plate, 2. pick up the plate, 3. find a microwave, 4. open the microwave, 5. put down the plate,
     ↪ 6. close the microwave, 7. turn on the microwave, 8. turn off the microwave, 9. open the microwave,
     ↪ 10. find a plate, 11. pick up the plate, 12. close the microwave, 13. find a dining table, 14. put down
     ↪ the plate, 15. done.
Human: Place a microwaved potato in the sink.
Robot: 1. find a potato, 2. pick up the potato, 3. find a microwave, 4. open the microwave, 5. put down the
     ↪ potato, 6. close the microwave, 7. turn on the microwave, 8. turn off the microwave, 9. open the
     ↪ microwave, 10. find a potato, 11. pick up the potato, 12. close the microwave, 13. find a sink, 14. put
     ↪  down the potato, 15. done.
Human: Moving a bowl to the shelf.
Robot: 1. find a cabinet, 2. open the cabinet, 3. find a cup, 4. pick up the cup, 5. close the cabinet, 6. find a
     ↪ microwave, 7. open the microwave, 8. put down the cup, 9. close the microwave, 10. turn on the
     ↪ microwave, 11. turn off the microwave, 12. open the microwave, 13. find a cup, 14. pick up the cup,
     ↪ 15. close the microwave, 16. find a shelf, 17. put down the cup, 18. done.
Human: Place a cleaned spatula on a dining table.

Robot: 1. find a spatula, 2. pick up the spatula, 3. find a sink, 4. put down the spatula, 5. find a faucet, 6. turn
   ↪ on the faucet, 7. turn off the faucet, 8. find a spatula, 9. pick up the spatula, 10. find a dining table,
   ↪ 11. put down the spatula, 12. done.
Human: Place the rinsed off apple on the brown kitchen table.
Robot: 1. find an apple, 2. pick up the apple, 3. find a sink, 4. put down the apple, 5. find a faucet, 6. turn on
   ↪ the faucet, 7. turn off the faucet, 8. find an apple, 9. pick up the apple, 10. find a dining table, 11. put
   ↪ down the apple, 12. done.
Human: Put a clean spatula on the counter.
Robot: 1. find a spatula, 2. pick up the spatula, 3. find a sink, 4. put down the spatula, 5. find a faucet, 6. turn
   ↪ on the faucet, 7. turn off the faucet, 8. find a spatula, 9. pick up the spatula, 10. find a counter top,
   ↪ 11. put down the spatula, 12. done.

Listing 2: Prompt for WAH-NL experiments when 10 in-context examples are provided (2 examples per task type). Subsets of the examples below are used when we use fewer examples, like 5, in the base experiments.

Robot: Hi there, I'm a robot operating in a home.
Robot: You can ask me to do various tasks and I'll tell you the sequence of actions I would do to accomplish
   ↪ your task.
Human: Please put 1 cutlery fork, 1 water glass and 1 plate in the dishwasher and turn the dishwasher on
Robot: 1. walk to dishwasher, 2. open dishwasher, 3. walk to cutlery fork, 4. grab cutlery fork, 5. walk to
   ↪ dishwasher, 6. put cutlery fork in dishwasher, 7. walk to water glass, 8. grab water glass, 9. walk to
   ↪ dishwasher, 10. put water glass in dishwasher, 11. walk to plate, 12. grab plate, 13. walk to
   ↪ dishwasher, 14. put plate in dishwasher, 15. close dishwasher, 16. switch on dishwasher, 17. done.
Human: store the pancake, the glass of wine, the cupcake and the glass of juice in the fridge
Robot: 1. walk to fridge, 2. open fridge, 3. walk to pancake, 4. grab pancake, 5. walk to fridge, 6. put pancake
   ↪ in fridge, 7. walk to wine, 8. grab wine, 9. walk to fridge, 10. put wine in fridge, 11. walk to cupcake
   ↪ , 12. grab cupcake, 13. walk to fridge, 14. put cupcake in fridge, 15. walk to juice, 16. grab juice, 17.
   ↪ walk to fridge, 18. put juice in fridge, 19. close fridge, 20. done.
Human: Take a pudding, an apple, a cupcake, and the juice and put them on the coffee table.
Robot: 1. walk to pudding, 2. grab pudding, 3. walk to coffee table, 4. put pudding on coffee table, 5. walk to
   ↪ apple, 6. grab apple, 7. walk to coffee table, 8. put apple on coffee table, 9. walk to cupcake, 10.
   ↪ grab cupcake, 11. walk to coffee table, 12. put cupcake on coffee table, 13. walk to juice, 14. grab
   ↪ juice, 15. walk to coffee table, 16. put juice on coffee table, 17. done.
Human: Find 1 coffee pot, 1 cupcake, 1 pudding and put on the kitchen table.
Robot: 1. walk to coffee pot, 2. grab coffee pot, 3. walk to kitchen table, 4. put coffee pot on kitchen table, 5.
   ↪ walk to cupcake, 6. grab cupcake, 7. walk to kitchen table, 8. put cupcake on kitchen table, 9. walk
   ↪ to pudding, 10. grab pudding, 11. walk to kitchen table, 12. put pudding on kitchen table, 13. done.
Human: I need a wine glass, water glass and a plate on the kitchen table
Robot: 1. walk to wine glass, 2. grab wine glass, 3. walk to kitchen table, 4. put wine glass on kitchen table, 5.
   ↪ walk to water glass, 6. grab water glass, 7. walk to kitchen table, 8. put water glass on kitchen table,
   ↪ 9. walk to plate, 10. grab plate, 11. walk to kitchen table, 12. put plate on kitchen table, 13. done.
Human: Add a fork and a plate to the dishwasher and switch it on.
Robot: 1. walk to dishwasher, 2. open dishwasher, 3. walk to cutlery fork, 4. grab cutlery fork, 5. walk to
   ↪ dishwasher, 6. put cutlery fork in dishwasher, 7. walk to plate, 8. grab plate, 9. walk to dishwasher,
   ↪ 10. put plate in dishwasher, 11. close dishwasher, 12. switch on dishwasher, 13. done.
Human: Could you put 1 apple, 1 pancake, 1 cupcake and 1 pudding in the fridge?
Robot: 1. walk to fridge, 2. open fridge, 3. walk to apple, 4. grab apple, 5. walk to fridge, 6. put apple in
   ↪ fridge, 7. walk to pancake, 8. grab pancake, 9. walk to fridge, 10. put pancake in fridge, 11. walk to
   ↪ cupcake, 12. grab cupcake, 13. walk to fridge, 14. put cupcake in fridge, 15. walk to pudding, 16.
   ↪ grab pudding, 17. walk to fridge, 18. put pudding in fridge, 19. close fridge, 20. done.
Human: Serve one wine on the coffe table
Robot: 1. walk to wine, 2. grab wine, 3. walk to coffee table, 4. put wine on coffee table, 5. done.
Human: Collect 1 apple, 1 coffee pot, and 1 juice and place them on the kitchen table
Robot: 1. walk to apple, 2. grab apple, 3. walk to kitchen table, 4. put apple on kitchen table, 5. walk to coffee
   ↪ pot, 6. grab coffee pot, 7. walk to kitchen table, 8. put coffee pot on kitchen table, 9. walk to juice,
   ↪ 10. grab juice, 11. walk to kitchen table, 12. put juice on kitchen table, 13. done.
Human: Put a plate, a fork and a water glass onto the kitchen table.
Robot: 1. walk to plate, 2. grab plate, 3. walk to kitchen table, 4. put plate on kitchen table, 5. walk to cutlery
   ↪ fork, 6. grab cutlery fork, 7. walk to kitchen table, 8. put cutlery fork on kitchen table, 9. walk to
   ↪ water glass, 10. grab water glass, 11. walk to kitchen table, 12. put water glass on kitchen table, 13.
   ↪ done.

Listing 3: Part of the prompt used in the replanning experiment. We used three additional replanning in-context examples, starting on the line following the prompt for ALFRED. The three in-context examples for replanning take one example from the prompt for ALFRED and add feedback messages for three different actions.

```
Robot: I can replan the action to perform successfully when the action fails with the message this action failed
    ↪ . If I generate the wrong plan, I'll make sure the command succeeds by replanning.
Human: Place the rinsed off apple on the brown kitchen table.
Robot: 1. find an apple, 2. pick up the apple, 3. find a sink, 4. put down the apple, 5. find a faucet, 6. turn on
    ↪ the faucet, 7. turn off the faucet, 8. find an apple, 9. pick up the apple, 10. find a dining table, 11. put
    ↪ down the apple (this action failed: put down failed), 12. find an apple, 13. pick up the apple, 14. find
    ↪ a dining table, 15. put down the apple, 16. done.
Human: Place the rinsed off apple on the brown kitchen table.
Robot: 1. find an apple, 2. pick up the apple, 3. find a sink, 4. put down the apple, 5. find a faucet, 6. turn on
    ↪ the faucet, 7. turn off the faucet, 8. find an apple, 9. find a dining table, 10. put down the apple (this
    ↪ action failed: Robot is not holding any object), 11. find an apple, 12. pick up the apple, 13. find a
    ↪ dining table, 14. put down the apple 15. done.
Human: Place the rinsed off apple on the brown kitchen table.
Robot: 1. find an apple, 2. pick up the apple (this action failed: Apple is not visible because it is in fridge), 3.
    ↪ find a fridge, 4. open the fridge, 5. find an apple, 6. pick up the apple, 7. find a sink, 8. put down the
    ↪ apple, 9. find a faucet, 10. turn on the faucet, 11. turn off the faucet, 12. find an apple, 13. pick up
    ↪ the apple, 14. find a dining table, 15. put down the apple, 16. done.
```

In replanning experiment, all actions had failure feedback messages. We describes three of them used in the prompt. The first feedback message is *"(this action failed: put down failed)"* when the put down action was failed because a receptacle was not found properly or because of physics-based simulation error. The second is the feedback message *"(this action failed: Robot is not holding any object)"* from the put down action, which failed to put down because the agent was not holding any object. Third, there is the feedback message *"(this action failed: [Object] is not visible because it is in [Container])"*, which is the feedback message from the pick up action when the object is not visible because the object is inside a container. The other feedback messages can be found in the supplementary code.

## G   DETAILS ON FINE-TUNED PLANNERS

We fine-tuned LLaMA 1 models using the ALFRED *train* set (Shridhar et al., 2020), which contains 17,468 pairs of user instructions and corresponding ground-truth plans. The dataset was formatted in accordance with the Alpaca instruction-following dataset (Taori et al., 2023) for fine-tuning, but utilizing a specialized prompt template, we aligned the final input for the fine-tuning process with the main prompt used in the baseline planner. A sample of this dataset is presented in Listing 4.

Listing 4: An excerpt from the fine-tuning dataset in the ALFRED domain

```
{
    ''instruction'': ''\nRobot: Hi there, I'm a robot operating in a home. \nRobot: You can ask me to do
        ↪ various tasks and I'll tell you the sequence of actions I would do to accomplish your task. \
        ↪ nHuman: Pick up the alarm clock and turn on the lamp.\nRobot:'',
    ''input'': '''',
    ''output'': ''1. find an alarm clock, 2. pick up the alarm clock, 3. find a desk lamp, 4. turn on the desk
        ↪ lamp, 5. done.''
},
{
    ''instruction'': ''\nRobot: Hi there, I'm a robot operating in a home. \nRobot: You can ask me to do
        ↪ various tasks and I'll tell you the sequence of actions I would do to accomplish your task. \
        ↪ nHuman: Carry an alarm clock while turning on a lamp\nRobot:'',
    ''input'': '''',
    ''output'': ''1. find an alarm clock, 2. pick up the alarm clock, 3. find a desk lamp, 4. turn on the desk
        ↪ lamp, 5. done.''
},
...
```

In the fine-tuning process, we utilized LoRA (Hu et al., 2021), with the hyper-parameters set as follows: the rank of LoRA modules was set to 16 (reduced to 8 for the 30B model due to GPU

memory constraints), the dropout rate was 0.1, and the number of epochs was 5. The benchmark conditions were consistent with those applied in the baseline planner tests, with the exception that we omitted in-context examples in the prompt since the fine-tuned planners had already been trained with 17,468 plan examples. Notably, in our experiment, the fine-tuned planners exhibited diminished performance when provided with in-context examples (for instance, LLaMA 1 30B showed the success rate of 44.59% in ALFRED *valid-unseen* set when there are six examples in the prompt, which is lower than the result without examples), a phenomenon demanding further exploration for comprehensive understanding.

Table 6 shows the results from the experiments of benchmarking fine-tuned planners. As shown in the table, our experiments indicate that the procedural knowledge does not transfer well to different task domains. Specifically, planners fine-tuned on the ALFRED domain underperformed compared to baseline few-shot planners in the WAH-NL domain.

Table 6: Planning success rates of planners fine-tuned on ALFRED. $\delta$ shows changes in success rates with and without fine tuning (pp refers percentage points).

| | ALFRED $\rightarrow$ ALFRED | | | | ALFRED $\rightarrow$ WAH-NL | |
| | *valid-seen tasks* | | *valid-unseen tasks* | | | |
| | Success(%) | $\delta$(pp) | Success(%) | $\delta$(pp) | Success(%) | $\delta$(pp) |
| LLaMA 1 7B | 32.21 | +26.44 | 37.39 | +31.62 | 3.70 | $-24.95$ |
| LLaMA 1 13B | 42.79 | +30.29 | 45.05 | +32.55 | 2.28 | $-21.42$ |
| LLaMA 1 30B | 57.69 | +44.03 | 60.08 | +46.42 | 10.38 | $-21.84$ |

The statistics of successful plans according to task types in ALFRED are shown in Table 7. Fine-tuned planners produced a more uniform distribution of successful plans across different task types.

Table 7: Number of successful plans by ALFRED task type. LLaMA 1 models were used and we reports results of the baseline and fine-tuning experiments for ALFRED *valid-seen* tasks.

| Task Type | | Baseline | | | Fine-tuned | | |
| (number of samples) | Sample Instruction | 7B | 13B | 30B | 7B | 13B | 30B |
| --- | --- | --- | --- | --- | --- | --- | --- |
| *Pick & Place* (49) | Move a book from a desk to a sofa. | 3 | 6 | 4 | 19 | 28 | 28 |
| *Stack & Place* (33) | Put a bowl with the watch in it on the shelf. | 0 | 0 | 0 | 11 | 10 | 16 |
| *Cool & Place* (39) | Place a cold potato slice in the sink. | 4 | 6 | 6 | 15 | 22 | 26 |
| *Clean & Place* (35) | Put a clean sponge on a metal rack. | 2 | 10 | 8 | 6 | 10 | 17 |
| *Heat & Place* (34) | Put a heated mug on the small black table, left of the tomato. | 0 | 1 | 7 | 13 | 10 | 24 |
| *Examine in Light* (15) | Turn on the tall lamp in the corner while carrying a white cushion. | 3 | 3 | 3 | 3 | 9 | 9 |

We also fine-tuned LLaMa 1 models on the WAH-NL *train* set. We utilized a single instruction per task, resulting in 250 pairs of user instructions and corresponding ground-truth plans from a total of 416 instructions. Consistent with our fine-tuning experiments on the ALFRED dataset, the same hyper-parameters and conditions were employed. The only difference was the inclusion of five in-context examples in the test prompt, addressing the limited number of WAH-NL *train* set and their impact on performance. Table 8 summarizes the results. While the LLaMA 1 7B model demonstrated a performance decrease of 7.34 percentage points, the larger LLaMA 1 13B and 30B models exhibited performance increases of 5.77 and 1.56 percentage points, respectively. These

results suggest that fine-tuning can enhance the performance of task planners, particularly in larger-scale language models, even with a constrained number of training examples.

Table 8: Subgoal success rates of planners fine-tuned on WAH-NL. $\delta$ shows changes in success rates with and without fine tuning (pp refers percentage points).

| | WAH-NL $\to$ WAH-NL | |
|---|---|---|
| | Success(%) | $\delta$(pp) |
| LLaMA 1 7B | 21.36 | $-7.34$ |
| LLaMA 1 13B | 29.47 | $+5.77$ |
| LLaMA 1 30B | 33.78 | $+1.56$ |

## H    ADDITIONAL PLANNING RESULTS

Success and failure results on the ALFRED dataset (Figure 8 and 9) and on the WAH-NL dataset (Figure 10 and 11). Figure 12 shows a replanning example described in Section 6.2.

Instruction: Move a book from a desk to a sofa.

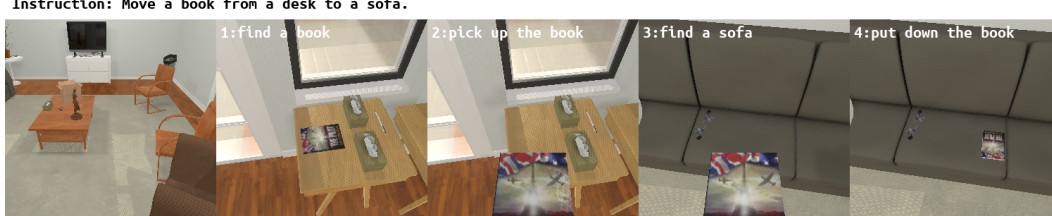

Instruction: Put the plunger into the bathroom cabinet

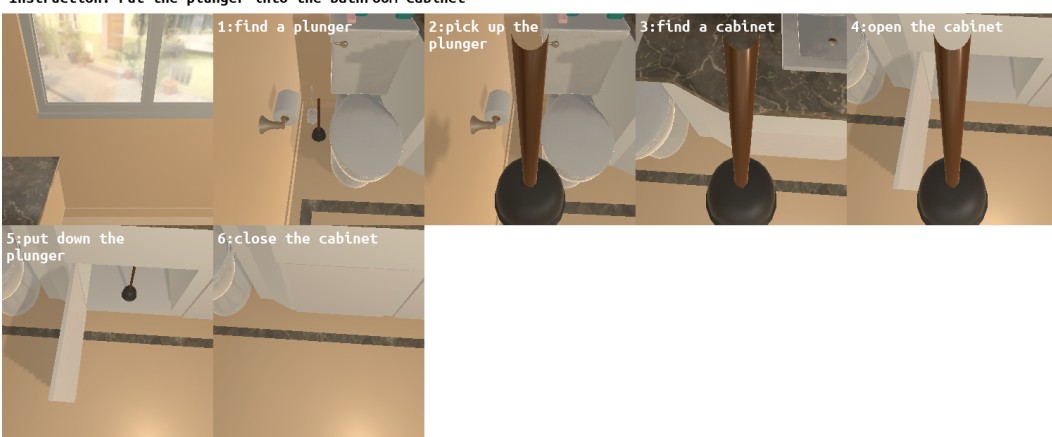

Instruction: drop a clean mug on the coffee maker

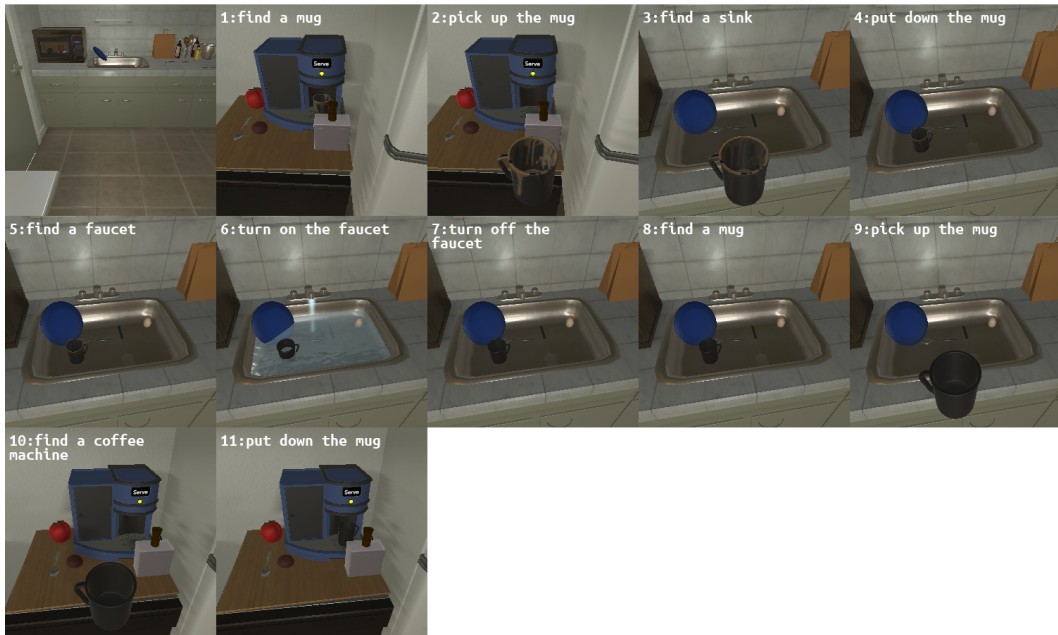

Figure 8: Success cases on ALFRED dataset. Results of the baseline planner with GPT-3 175B.

Instruction: Put a bar of soap on the back of the toilet.

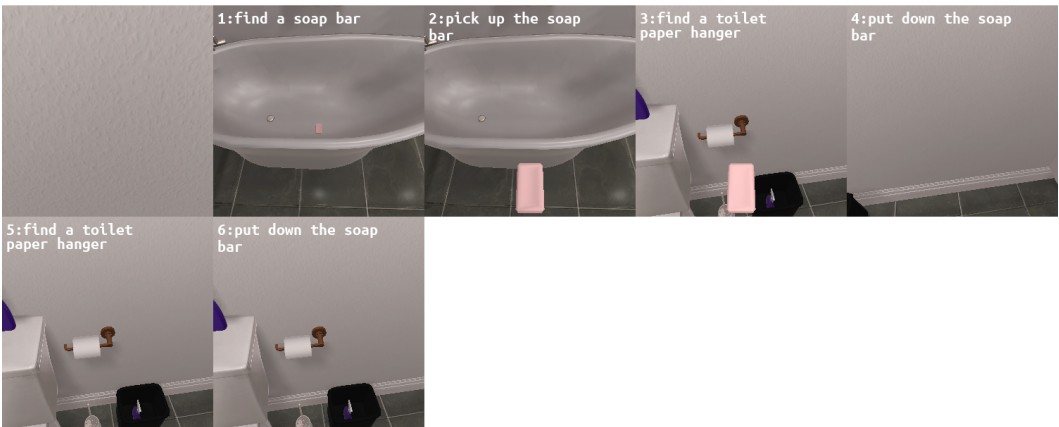

Instruction: Put the mug with fork on the round table.

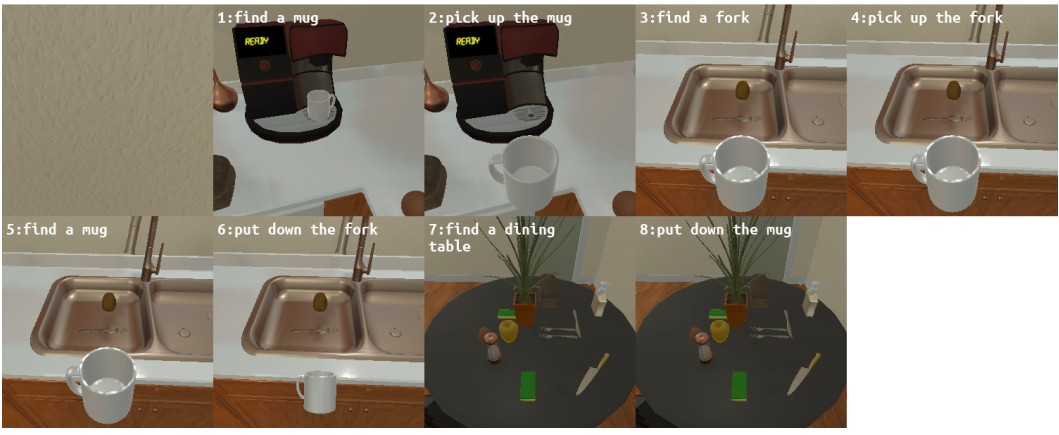

Instruction: trash a cooked potato slice

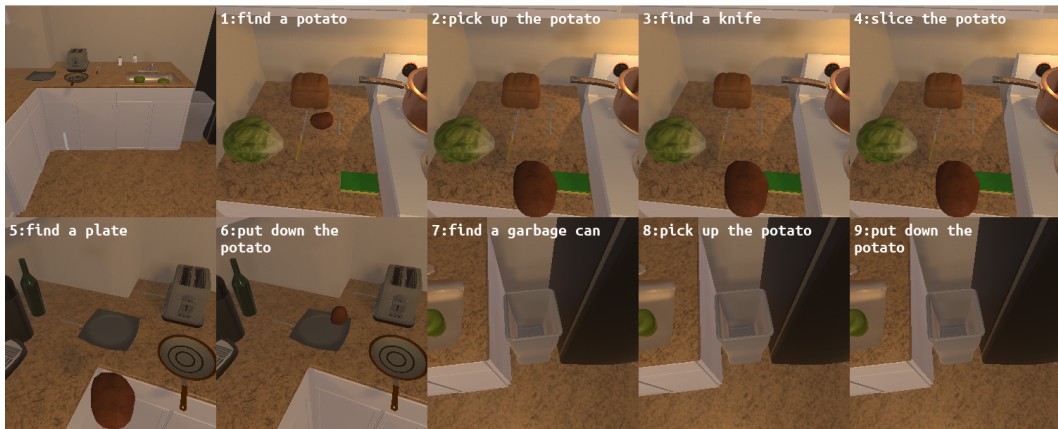

Figure 9: Failure cases on ALFRED dataset. Results of the baseline planner with GPT-3 175B.

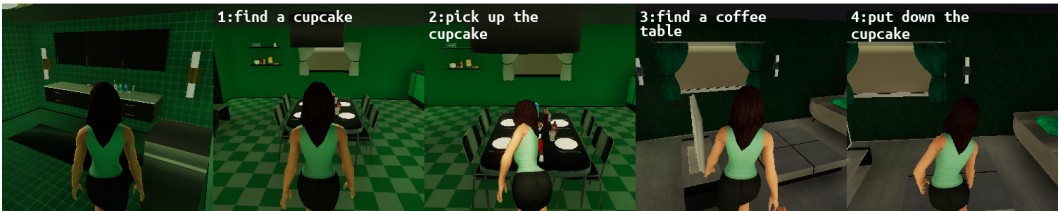

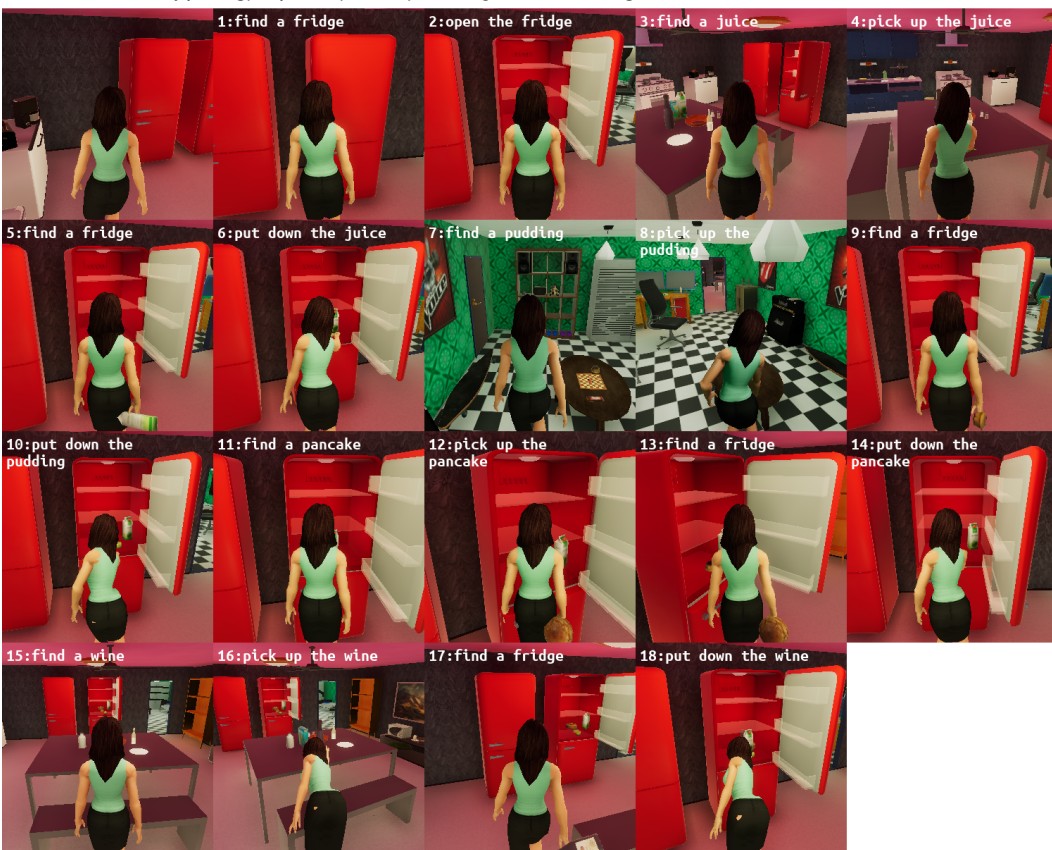

Figure 10: Success cases on WAH-NL dataset. Results of the baseline planner with GPT-3 175B.

Instruction: On the kitchen table are 1 apple, 1 wine, 1 cupcake and 1 pudding.

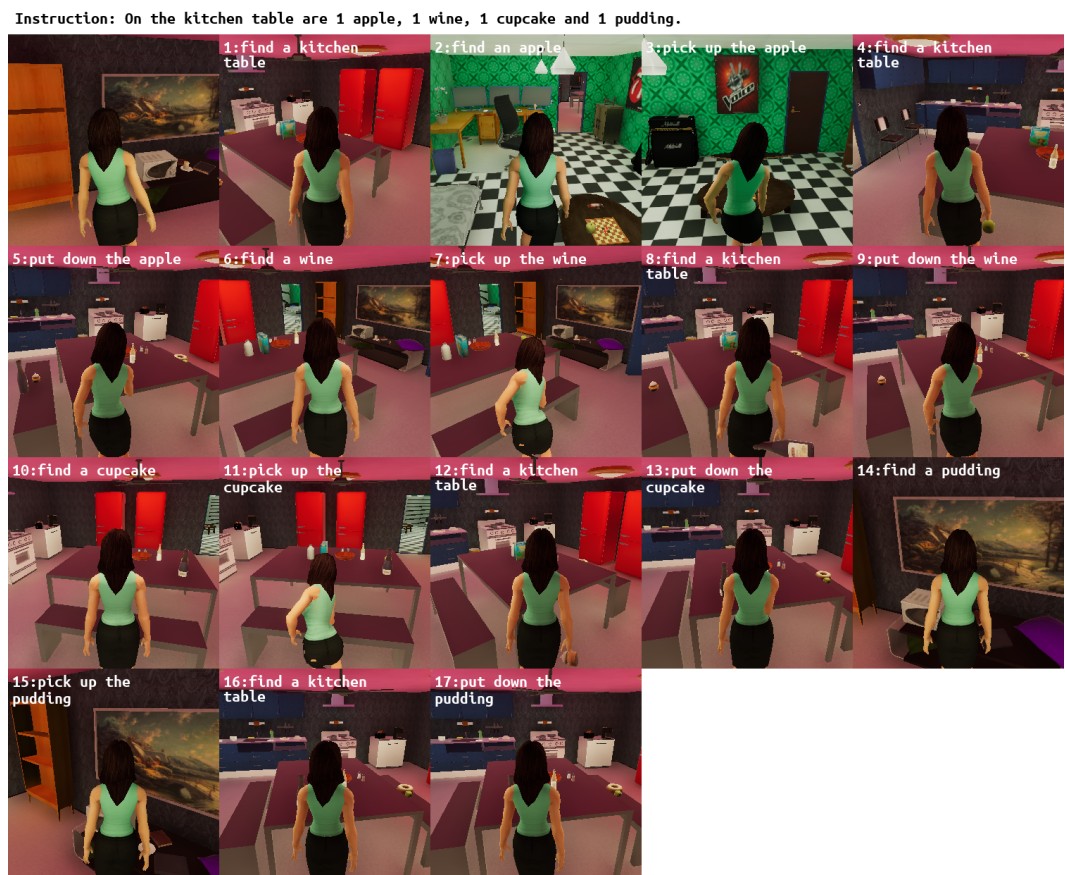

Instruction: Please serve a water glass on the kitchen table

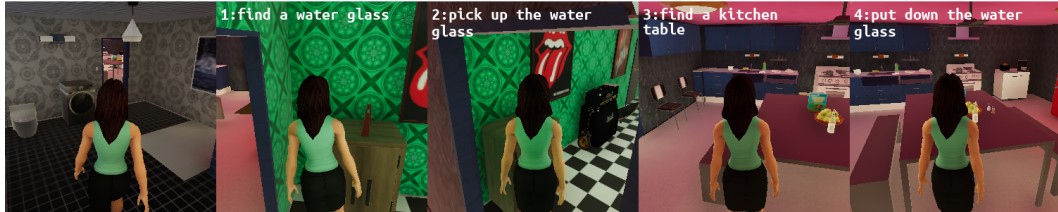

Figure 11: Failure cases on WAH-NL dataset. Results of the baseline planner with GPT-3 175B.

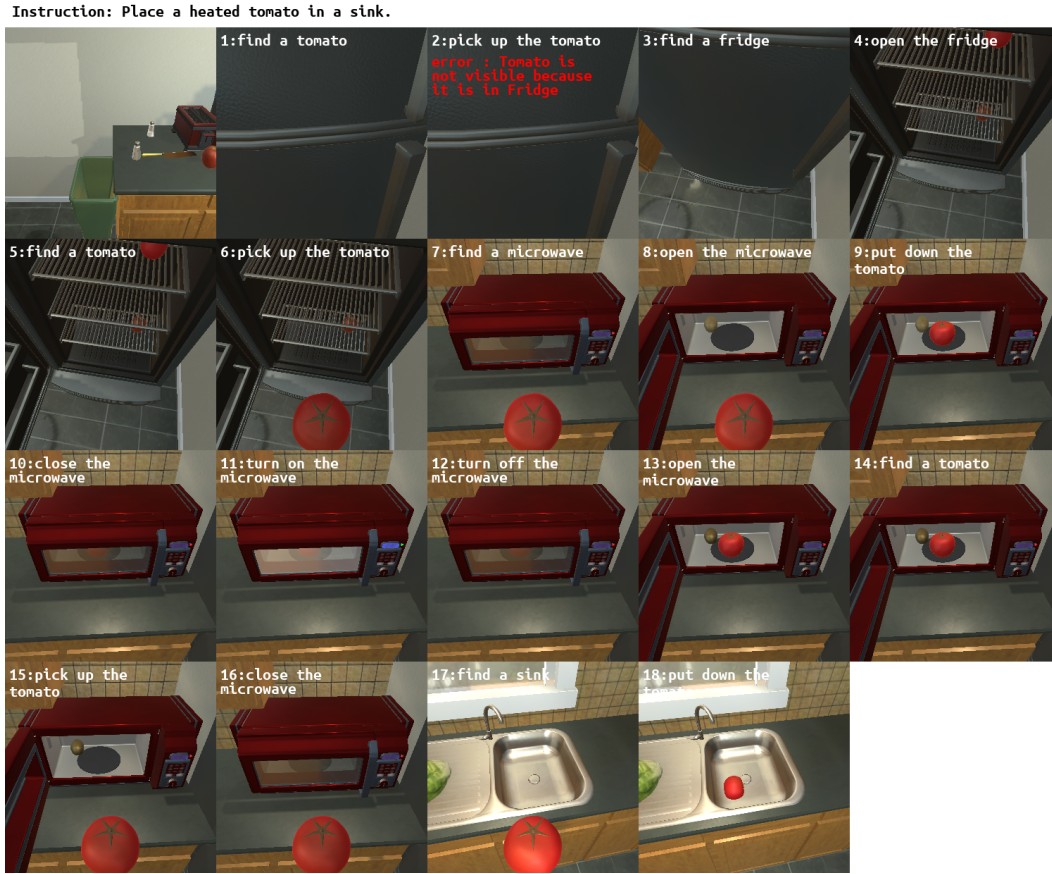

Figure 12: Success cases of replanning on the ALFRED dataset using LLaMA 2 70B. (Top) Success case when replanning in pick up action. (Bottom) Success case when replanning in put down action.

