# OpenReview forum: "LoTa-Bench: Benchmarking Language-oriented Task Planners for Embodied Agents"
_ICLR.cc/2024/Conference — ICLR 2024 poster_

### Official Review · Reviewer_mB86 · 2023-10-19

**Soundness:** 3 good
**Presentation:** 3 good
**Contribution:** 2 fair
**Rating:** 6
**Confidence:** 4

**Summary:**

The paper proposes LoTa-Bench, an evaluation framework for the planning capability of LLMs for embodied agents.
The proposed benchmark utilizes two existing benchmarks, ALFRED and WAH, that require agents to plan a sequence of actions to satisfy desired goal conditions given natural language instructions.
It follows the evaluation protocol of ALFRED and WAH.
The paper compares various large language models for planning. It conducts some experiments to show possible extensions: 1) in-context example selection, 2) replanning based on feedback, and 3) fine-tuning LLMs on downstream tasks.

**Strengths:**

- The paper is generally written well and easy to follow.
- The paper provides an important framework for the evaluation of the planning ability of LLMs which have been widely used for the planning of embodied agents.
- Focusing on planning by language by some assumptions (e.g., oracle information of objects for visual navigation) sounds reasonable.
- The paper provides a useful extensive analysis of the planning capabilities of existing LLMs for the proposed benchmark.

**Weaknesses:**

- I appreciate the proposed benchmark for LLMs' planning but some of the proposed possible extensions on this benchmark are somewhat expected. For example, fine-tuning a pretrained model (here, LLMs) to downstream tasks is a widely used approach. And for replanning by feedback, while a larger LLM is improved with the replanning, a smaller one is not, implying that we need some large LLMs, but it is also quite expected that a larger model may yield better performance and a smaller one.
- The feedback for replanning is based on metadata (i.e., oracle information about what is "wrong") of the simulated environments but this may not be always available beyond the benchmark for other downstream tasks (e.g., some tasks might not be able to provide why agents fail at actions). Can this direction be also useful for those cases without the oracle information?
- The authors conduct experiments based on a subset (30%) of the original datasets but do not provide some significance tests. It might be better to include them in the results (e.g., Figure 2) with different subsets.

\* Minor
 - The used font looks a bit different from the one used in the original ICLR format. It might be better to revert to the original font.

**Questions:**

See weaknesses above.

---

> ### Author Response · Authors · 2023-11-19
>
> Thank you for your thorough review and constructive feedback. We are grateful for the positive remarks on the clarity of the paper and the significance of the proposed framework for evaluating language models' planning capabilities. We respond to your comments on the weaknesses.
>
> > **Some of the proposed possible extensions on this benchmark are somewhat expected.**
>
> As you have mentioned, the two extensions we proposed do exhibit somewhat predictable outcomes. However, the significant benefit of our LoTa-Bench lies in its ability to quantitatively measure the effectiveness of fine-tuning, a widely used approach. This capability is valuable as it provides a concrete benchmark to evaluate the extent of improvement that fine-tuning can offer.
>
> In the case of replanning by feedback, our benchmark also offers valuable insight. The performance variations when using NL feedback highlight that different types of information can significantly influence outcomes. This aspect of our work facilitates easy testing of these variables. It provides a robust test environment that can shape the direction of future research in LLM-based task planning. We believe that our LoTa-Bench will encourage the community to experiment with their ideas, and this will bring interesting results.
>
> > **The feedback for replanning is based on metadata ... but this may not be always available beyond the benchmark for other downstream tasks ... Can this direction be also useful for those cases without the oracle information?**
>
> In Section 5.2, we conducted experiments on feedback and replanning. As you mentioned, we utilized oracle information to articulate the reasons for failure in the current step in natural language, which is then fed into an LLM-based planner for replanning. These experiments primarily investigated the impact of NL feedback on planning performance, under the assumption that reasons for failure are accessible. However, in scenarios where oracle information is unavailable, our approach could potentially infer the causes of step failures using sensor data from embodied agents. For instance, feedback like “(this action failed: Robot is not holding any object)” could be derived by checking if the robot’s hand sensors detect an object. Similarly, “(this action failed: put down failed)” could be inferred using vision data by object detection. This possibility represents a separate research direction. Our paper primarily focuses on evaluating the effectiveness of NL feedback when it is available, demonstrating its potential to enhance high-level planning performance.
>
> > **The authors conduct experiments based on a subset (30%) of the original datasets but do not provide some significance tests. It might be better to include them in the results (e.g., Figure 2) with different subsets.**
>
> Although we randomly selected a 30% subset, it might have some biases in selection. We fully agree with you that it would be better to include results with different subsets. We conducted additional experiments with three other subsets. Due to the limited time of this discussion period, we only included results for three models of GPT-Neo, LLaMA-1, and LLaMA-2 except for the large ones having more than 60B parameters. Our code was ready for this experiment; we only changed the random seed for subset selection on line 78 in *src/alfred/alfred_evaluator.py* in the supplementary code.
>
> The figure linked below shows the results with different subsets.
>
> https://figshare.com/s/557079abb37b1712d074
>
> The bold lines are the same as the previous results in Figure 2. The thin lines represent additional results with different subsets. The general trends that 1) bigger models work better and 2) LLaMA series are better than GPT-Neo are the same. However, we found some differences in the results. For example, the success rates range widely from 7.21 to 12.5 for GPT-NeoX-20B according to the selection of subsets. And LLaMA-1 worked better than LLaMA-2 for some subsets. As we found subset selection affect results to some extent, we will also include full set (using 100\% samples) results with all the models in the final paper (if we have the opportunity to prepare the final paper). Specifically, we will include two figures for the subset and the full set instead of Figure 2 (a).
>
> We would like to mention that reporting results for a subset is still valuable since many researchers do not have enough computing resources, so we think it would be better to maintain a 30% subset as well so that it serves as a kind of small set for quick or affordable experiments.
>
> This subset experiment is only for ALFRED. We already used the full set for WAH-NL.

---

> ### Author Response · Authors · 2023-11-19
>
> (continued)
>
> > **The used font looks a bit different from the one used in the original ICLR format. It might be better to revert to the original font.**
>
> Thank you for pointing this out. We used XeLaTeX for PDF compiling, and it gave slightly different fonts than the pdfLaTeX compiler. We corrected the compiling options and will use the corrected version when we upload the revised draft. It was not intentional, and the page count was the same.
>
> ---
>
> We appreciate your valuable feedback, and we believe that addressing these points will contribute to the overall improvement of the paper. If you have any further suggestions or remaining concerns, please do not hesitate to communicate them.

---

> > ### Comment · Reviewer_mB86 · 2023-11-22
> > **Official Comment by Reviewer mB86**
> >
> > I thank the authors for their detailed response.
> > The response addressed my concerns and thus I would like to keep my original positive rating.

---

### Official Review · Reviewer_DjjY · 2023-10-31

**Soundness:** 3 good
**Presentation:** 3 good
**Contribution:** 2 fair
**Rating:** 6
**Confidence:** 4

**Summary:**

The paper proposed the development of a benchmark to measure the effectiveness of LLM-based task planners in language-related settings (particularly where the system receives natural language instructions). The proposed benchmark consists of pairs of datasets and simulators, where the dataset includes example instructions and environment information, and the simulator allows the plan generated by the planner to be evaluated. They propose a baseline planner that selects the highest probability skill from a set of possible skills, given the relevant information. The base planner is evaluated on the two proposed dataset/simulator pairs, and they also perform additional experiments to evaluate factors like replanning, finetuning, and example selection.

**Strengths:**

The clear strength of the paper is the fact that this is a problem that is getting a lot of attention. As discussed by the authors, there aren’t quite too many benchmarks of this specific quality, particularly the presence of simulators ( though I am not completely sure if this aspect is being fully utilized here). In addition to the basic evaluation, I found the additional experiments quite insightful and useful, especially the ones related to the example selection.

**Weaknesses:**

NL to API Component: I am a little bit confused about the need for setting up an NL to API component for the current system. Why can’t the skills provided to the LLM consist of directly executable API calls or actions? Is the concern here that the API calls would be too abstract and arbitrary to be connected to the actual instruction? Can’t you get around it by using meaningful action names (as is the case in classical symbolic planning settings [1]) or provide a description of each action in natural language as part of the environment context?

Failure Reasons: I wish there were a more detailed analysis of how the plans failed and the main results were not reported in terms of mere success rates. Some potential analyses the authors should consider here include things like checking if there were any phenomena akin to invalid preconditions, i.e., a selected low-level controller needed certain state constraints to be met before it was applicable. Similarly, how close were the failed plans to valid plans? Could you have turned them into valid plans by appending a few more actions? Were they moving the environment state closer to the goal state? Was the goal state achieved as an intermediate result and then changed by some future actions?

Lack of tests on GPT-4: One shortcoming of the current set of experiments is that it is missing any results that make use of GPT-4 as a planner, which is widely recognized as being one of the most powerful LLMs that are currently available. The reason cited is the unavailability of mechanisms to identify the probability associated with each possible next token. However, as I understand it, the basic planner uses a greedy approach to select the next most likely token. Each selected token is appended to the prompt, and the next token is selected. Isn’t this similar to the basic autoregressive mechanism used by all LLMs, including GPT-4, especially when using low temperatures? You could have forced GPT-4 to only select tokens from the pool of potential skills by careful prompt engineering and some filtering. Even if the mechanism weren’t exactly the same, this would have provided a useful point of comparison on how the state-of-the-art LLM models fair in these benchmark tasks.

[1] Haslum, Patrik, et al. An introduction to the planning domain definition language. Vol. 13. San Rafael, California: Morgan & Claypool, 2019.

**Questions:**

Please answer the questions raised under the weaknesses.

---

> ### Author Response · Authors · 2023-11-21
>
> Thank you for your thorough review and constructive feedback. Your feedback provides valuable insights that we address in this comment.
>
> > **NL to API Component**
>
> In both AI2THOR and VirtualHome simulators, directly executable API calls require object instances. For example, AI2THOR uses a format like {action=“PickupObject”, objectId=1}, and VirtualHome might use “[grab] <coffeepot> (101)”. We could design the skill set to include object IDs, such as S = {“[grab] <coffeepot> (101)”, …, “[open] <fridge> (10)”}, enabling direct API calls. However, this method would substantially increase the skill set size, resulting in higher computational costs.
>
> To mitigate these issues, we formulated each skill in natural language, without object instances. The NL to API component is responsible for translating these NL skills into executable API calls, selecting the nearest object ID for execution. This approach aligns with your suggestion of using “meaningful action names.”
>
> Furthermore, employing NL skills also assists in avoiding simulator dependency, a crucial factor in maintaining system flexibility and generalizability. This strategy enables our system to adapt seamlessly across different simulation environments.
>
> > **Failure Reasons**
>
> We have analyzed the failure cases in more detail based on your suggestion. We specifically examined the detailed reasons for the failures in the ALFRED results of GPT-3 (text-davinci-003) model, which showed the highest performance. For 162 failure cases, the authors manually categorized the failure reasons. We classified the failures into six categories.
>
> * 1) Failure of action planning (e.g., performing 'Pick' instead of 'Slice' at the step where a tomato need to be sliced)
> * 2) Failure of object selection (e.g., grabbing a pan instead of a pot)
> * 3) Failure due to absence of visual grounding (e.g., attempting to grab an object inside a drawer without opening it, failure to distinguish instances when multiple objects of the same kind are present)
> * 4) Lack of physical understanding (e.g., failure to put down an object because there are already other objects on the table)
> * 5) Failure in understanding user instructions (e.g., failure to differentiate whether the user-specified 'Lamp' refers to a desk lamp or a floor lamp)
> * 6) Ambiguous or incorrect user instructions (e.g., referring to 'Glass' in an instruction but the actual goal condition is 'Cup').
>
> The results of the failure analysis are presented in the table below.
>
> | Failure category | Number of failures |
> |-------------|---------------------|
> | 1. Failure of action planning                        |  46  |
> | 2. Failure of object selection                    |  51  |
> | 3. Failure due to absence of visual grounding     |  21  |
> | 4. Lack of physical understanding                 |  15  |
> | 5. Failure in understanding user instructions     |  10  |
> | 6. Ambiguous or incorrect user instructions       |  19  |
>
>
> This may serve as valuable information for understanding the causes of failures in the current benchmark. We found that the main reason of failure is high-level planning failures (1 and 2), which is about 60% of the total failure.
> Visual grounding and physical grounding (3 and 4) also matters and it is important future research direction of considering context grounding in high-level planning as we discussed in the conclusion and limitation section. Task instructions from human users might also be incomplete, unclear, or even incorrect (5 and 6). It would open a new research direction in which robot agents interactively request clarification of an ambiguous goal.

---

> ### Author Response · Authors · 2023-11-21
>
> (continued)
>
> > **Lack of tests on GPT-4**
>
> As OpenAI provides only chat-style APIs for GPT-4 [1] unlike other base models such as GPT-3 (specifically, text-davinci-003), we were unable to directly compare GPT-4 in the same configuration. However, as you suggested, we aimed to assess GPT-4's performance, the state-of-the-art model, even with some modified experimental configurations. The following describes the configuration details and results of the GPT-4 experiments that we conducted.
>
> **GPT-4 Experiment Setting**
>
> 1. Prompt
>
>     GPT-4 API has three agent roles of “system,” “user,” and “assistant.” We modified the original prompt (Listing 1 in the supplementary material) to fit these roles as follows. It worked like a prompt that gives the context of a problem and the role of the robot agent. The number of examples provided for in-context learning was the same. Examples, robot skills, and task query in square brackets [] were replaced with actual texts according to the test sample and environment.
>
>     ```
>     <System Role>
>     You are a robot operating in a home. A human user can ask you to do various tasks and you are supposed to tell the sequence of actions you would do to accomplish your task.
>
>     <User Role>
>     Examples of human instructions and possible your (robot) answers:
>     [The same in-context-learning examples are provided]
>
>     Now please answer the sequence of actions for the input instruction.
>     You should use one of actions of this list: [List of robot skills provided]
>     List the actions with comma separator.
>
>     Input user instruction: [Task query]
>     ```
>
> 2. Parsing a response
>
>     An assistant (robot) agent generates action sequences for the given user instruction. Due to the unavailability of Guidance’s token selection mechanism [2] for GPT-4, we provided an admissible action list in the prompt to circumvent the issue.
>
>     GPT-4 generated the answer for the given user instruction, and we extracted each step from the answer text. There was no additional post-processing tailored to GPT-4 (steps not in the provided list of the skills would not be executed correctly).
>
>
> **Results**
>
> | Model | ALFRED | WAH-NL |
> |-------------|---------------------|---------------------|
> | GPT-3 (text-davinci-003)  |  21.36%  |  40.82%  |
> | GPT-4                     |  40.38%  |   34.17%   |
> | GPT-3.5-Turbo             |  10.58%  |   28.82%   |
>
> We tested GPT-4 and GPT-3.5-Turbo. GPT-3.5-Turbo showed lower success rates for both ALFRED and WAH-NL compared to GPT-3. GPT-4 performed well in ALFRED, showing a 40.38% success rate, a 19% improvement over GPT-3. However, in WAH-NL, GPT-4 demonstrated a lower success rate of 34.17% compared to GPT-3.
>
> Despite the limitations of the current setup, GPT-4's performance notably surpassed that of GPT-3 in ALFRED. This observation leads us to anticipate that the incorporation of Guidance’s token selection mechanism [2] for GPT-4 could yield more impressive planning capabilities.
>
>
> [1] https://platform.openai.com/docs/guides/text-generation/chat-completions-api \
> [2] https://github.com/guidance-ai/guidance
>
> ---
>
> We appreciate your valuable feedback, and we believe that addressing these points will contribute to the overall improvement of the paper. We will revise the paper to include the failure analysis and the GPT-4 results. If you have any further suggestions or remaining concerns, please do not hesitate to communicate them.

---

> ### Author Response · Authors · 2023-11-23
> **Looking Forward to Your Feedback**
>
> Dear Reviewer DjjY,
>
> Thank you again for your considerable comments for our paper. We hope that our rebuttal has addressed your concerns. As the ICLR discussion deadline approaches, we kindly request your feedback on our submitted response. If you have any further questions or require additional clarification, please let us know.
>
> Thank you very much for your time!
>
> Best wishes,
>
> The Authors

---

### Official Review · Reviewer_JeGM · 2023-11-01

**Soundness:** 3 good
**Presentation:** 2 fair
**Contribution:** 3 good
**Rating:** 6
**Confidence:** 4

**Summary:**

The paper presents LoTa-Bench, a benchmark for evaluating language task planners in home-service robots, using two datasets/simulators: ALFRED with AI2-THOR and Watch-And-Help with VirtualHome. It contributes a benchmarking suite, extensive testing of baseline planners with different models and prompts, exploration of planner enhancements, and the release of the benchmark code and an extended dataset to the public. The findings highlight the importance of model selection, in-context example strategies, replanning mechanisms, and model fine-tuning in improving planner performance.

**Strengths:**

1. **Innovative Benchmark Development**: The paper introduces LoTa-Bench, an innovative benchmark suite for language-oriented task planners. This benchmark is critical for the field as it allows for the automatic and reproducible evaluation of task planning, which is a significant step forward in embodied AI research.

2. **Comprehensive Experimental Analysis**: The authors provide extensive experimental results, which is a comprehensive demonstration of the benchmark's capabilities. They conduct experiments across various pre-trained models, demonstrating the benchmark's utility in evaluating the impact of model selection, in-context examples, replanning, and fine-tuning.

3. **Resource Contribution and Future Research Facilitation**: The paper contributes valuable resources to the research community, including the benchmark code and an extended dataset. Moreover, the findings and the benchmark itself are positioned to facilitate future research, potentially accelerating advancements in the field of language-oriented task planning for embodied agents.

**Weaknesses:**

1. **Decoupling of Planning Levels**: The research decouples high-level plans from low-level actions, focusing only on the former. This separation might not accurately reflect the complexities of end-to-end task planning where high-level decisions and low-level actions are interconnected.

2. **Lack of Visual Understanding**: The benchmark does not incorporate visual understanding, which is critical for low-level actions, especially in embodied AI where egocentric views play a significant role in interpreting and interacting with the environment.

3. **Domain Gap in Simulation**: There is a domain gap between the simulation environments used and the real world, leading to unrealistic assumptions. For example, the ALFRED simulator assumes an object is clean once it is placed in water, which may not always be true in real-world scenarios.

4. Another limitation of the paper is the absence of an in-depth investigation into several influential factors that affect the performance of task planners. These factors include the type and size of pre-trained language models, the number and strategy for selecting in-context examples, the ability of planners to replan based on natural language feedback, and the impact of fine-tuning the models​.  Addressing these factors is crucial for a thorough understanding of how to optimize task planners for real-world applications.

Minor one:
Missing references:
- On Grounded Planning for Embodied Tasks with Language Models
- Plan, Eliminate, and Track - Language Models are Good Teachers for Embodied Agents

**Questions:**

1. How does the decoupling of high-level plans from low-level actions in your benchmark reflect on the practical deployment of task planners? Could this lead to overlooking important dynamics that only emerge when both levels of planning are considered together?

2. Given that AI2THOR and VirtualHome lack diversity to reflect real-world environments, could you elaborate on your plans to enhance these simulators or to incorporate additional platforms that might offer more realistic and diverse scenarios?

3. How do you plan to address the domain gap between simulation and real-world application, and do you have strategies for dealing with unrealistic assumptions like the one mentioned in ALFRED about objects being cleaned once put into water?

4. Can you discuss the potential development directions for evaluation frameworks that can handle the complexity of LLM-based task planning beyond simple tabletop manipulation tasks, as mentioned in the current limitation of the field?

---

> ### Author Response · Authors · 2023-11-21
>
> Thank you for your valuable feedback. We appreciate your recognition of the strengths in our paper, especially the innovative benchmark development, comprehensive experimental analysis, and the valuable resources contributed to the research community. We would like to respond to your comments on the weaknesses and questions.
>
> > **[W1] Decoupling of Planning Levels**
>
> > **[Q1] How does the decoupling of high-level plans from low-level actions in your benchmark reflect on the practical deployment of task planners? Could this lead to overlooking important dynamics that only emerge when both levels of planning are considered together?**
>
> Decomposing complex long-horizon tasks into multiple levels of planning is a widely used strategy. Consistent with this approach, our paper decouples high-level plans from low-level controls as you mentioned. We intentionally focused on high-level planning, particularly employing an LLM-based method. Our approach offers enhanced generalizability compared to end-to-end methods, which typically require extensive, task-specific data collection and training [1, 2, 3]. Our methodology is inherently more flexible, capable of adjusting to new environments without the need for additional data collection or exhaustive training.
>
> Furthermore, we recognize the importance of ensuring that our high-level planning seamlessly integrates with low-level controllers through natural language. Thanks to recent advancements in object navigation [4] and language-conditioned manipulation [5, 6], the NL steps generated by LLM-based task planners can be effectively executed.
>
> [1] Learning Neuro-Symbolic Skills for Bilevel Planning \
> [2] Value Function Spaces: Skill-Centric State Abstractions for Long-Horizon Reasoning \
> [3] Hierarchical Foresight: Self-Supervised Learning of Long-Horizon Tasks via Visual Subgoal Generation \
> [4] A Survey of Embodied AI: From Simulators to Research Tasks \
> [5] BC-Z: Zero-Shot Task Generalization with Robotic Imitation Learning \
> [6] RT-1: Robotics Transformer for Real-World Control at Scale
>
> > **[W2] Lack of Visual Understanding**
>
> In response to this concern, we wish to clarify that our LoTa-Bench is strategically designed to concentrate on high-level task planning, distinct from vision-based low-level control. The benchmarks’s primary goal is to provide a clear and focused evaluation of language-oriented high-level task planning performance, thus intentionally minimizing potential confounding influences from low-level controls.
>
> However, we agree with your point regarding the importance of visual understanding. We are actively considering potential expansions to LoTa-Bench that would integrate visual information. This extension is designed not to exploring visual understanding for low-level control purposes but to enrich the capabilities of high-level planning. Potential directions include leveraging natural language feedback informed by visual information or integrating visual affordances into the planning process. Our objective with these future developments is to effectively bridge the gap between language-oriented task planning and visual perception.
>
> > **[W3] Domain Gap in Simulation**
>
> > **[Q3] How do you plan to address the domain gap between simulation and real-world application, and do you have strategies for dealing with unrealistic assumptions like the one mentioned in ALFRED about objects being cleaned once put into water?**
>
> As you pointed out, since LoTa-Bench evaluates planning performance in simulator environments, there is a domain gap with real world. The most prominent gap is ALFRED's assumption about cleaning. While other simulation tasks, such as picking and placing, heating using a microwave, and washing using a dishwasher, exhibit a smaller gap from real-world scenarios at a high-level planning stage.
>
> In response to this, one potential approach to expand the proposed benchmark is to enhance the AI2THOR simulator used in ALFRED to reduce the domain gap. It may become more realistic by adding conditions, such as requiring the target object to be touched with a sponge, particularly addressing unrealistic gaps like cleaning, and incorporating this into the object status update logic in the simulator. Additionally, cleaning using a dishwasher, as supported in WAH-NL, could also be added. For example, instead of simply placing a fork in the water, we can add steps, placing fork inside a dishwasher, close the dishwasher, and turn on the machine.

---

> ### Author Response · Authors · 2023-11-21
>
> (Continued)
> > **[W4] Another limitation of the paper is the absence of an in-depth investigation into several influential factors that affect the performance of task planners. ... Addressing these factors is crucial for a thorough understanding of how to optimize task planners for real-world applications.**
>
> Our paper indeed addresses each of the factors you have mentioned, using our proposed LoTa-Bench framework. The details are described in Section 5 and 6 of our manuscript.
>
> * **1) Impact of types and sizes of pre-trained language models**
>
>     In our work, we utilized the LoTa-Bench framework to evaluate and analyze the task planning performance of various pre-trained LLMs, including GPT [7], GPT-Neo series [8, 9, 10], OPT [11], MPT [12], LLaMA 1 [13], and LLaMA 2 [14], and their sizes. Our experimental results, summarized in Figure 2 of the paper, show that task planning performance generally increases with the size of the language model. Specifically, GPT-3 175B demonstrates the best performance on both ALFRED and WAH-NL. Further details are provided in:
>     * **Section 5.2**: “We evaluated the planning performance of the baseline planner described in Section 3. … Instruction- and chat-tuned models (dashed lines in Figure 2) did not perform better than their base models.” and **Figure 2**
>
> * **2) Influence of the number of in-context examples**
>
>     We explored the influence of the number of in-context examples using LLaMA 2 13B model. Our tests varied the number of examples from 0 to 30 in ALFRED and from 0 to 15 in WAH-NL. The results, depicted in Figure 3 of the paper, indicates that planning performance improves with an increased number of examples. More information can be found in:
>     * **Section 5.2**: “We have investigated the impact of the number of examples in prompt with LLaMA 2 13B model that supports a longer context length of 4096. … (with an average of 13.61 and a standard deviation of 3.22) for the LLaMA 2 13B model.” and **Figure 3**
>
> * **3) Strategy for selecting in-context examples**
>
>     We introduced three strategies (Random Sampling, Task-Specific Sampling, Semantic Similarity Sampling) for selecting in-context examples from the train set in Section 6.1 and investigated their impact on planning performance. Notably, Semantic Similarity Sampling significantly enhances planning performance. More details are available in Section 6.1 and Figure 5.
>
> * **4) Planners’ ability to replan based on natural language feedback**
>
>     In Section 6.2, we discuss the effect of feedback and replanning. We observed that providing NL feedback about the reasons for action failure aids task planners in replanning when the size of language model is large. Further insights are available in Section 6.1 and Table 1.
>
> * **5) Impact of fine-tuning on train set**
>
>     Our investigation into the impact of fine-tuning on the train set is discussed in Section 6.3. We employed LoRA [15] for fine-tuning LLMs, which resulted in significant performance improvements. This is elaborated upon in: Section 6.2 and Figure 6.
>
> We would like to emphasize the critical role of our LoTa-Bench framework in facilitating these comprehensive evaluations. Without such a benchmark suite, conducting in-depth analyses of task planner performance across the factors you have highlighted would be more challenging. Our framework not only enables these detailed evaluations but also provides substantial support in advancing the field of task planning.
>
> [7] Language Models are Few-Shot Learners \
> [8] GPT-Neo: Large Scale Autoregressive Language Modeling with Mesh-Tensorflow \
> [9] GPT-J-6B: A 6 Billion Parameter Autoregressive Language Model \
> [10] Gpt-neox-20b: An open-source autoregressive language model \
> [11] OPT: Open Pre-trained Transformer Language Models \
> [12] Introducing mpt-7b: A new standard for open-source, commercially usable llms \
> [13] LLaMA: Open and Efficient Foundation Language Models\
> [14] Llama 2: Open Foundation and Fine-Tuned Chat Models \
> [15] LoRA: Low-Rank Adaptation of Large Language Models
>
> > **[Q2] Given that AI2THOR and VirtualHome lack diversity to reflect real-world environments, could you elaborate on your plans to enhance these simulators or to incorporate additional platforms that might offer more realistic and diverse scenarios?**
>
> Our evaluation framework is designed to be extendable with different dataset and simulator pairs. For successful integration, the dataset must include natural language instructions and goal conditions, while the simulator should support high-level APIs. To enrich our framework with broader range of scenarios, we are actively considering the integration of additional platforms. Currently, we are examining the potential of incorporating Behavior-1K and OmniGibson simulator [16], or creating custom datasets within the NVIDIA Isaac Sim simulator.
>
> [16] BEHAVIOR-1K: A Benchmark for Embodied AI with 1,000 Everyday Activities and Realistic Simulation

---

> ### Author Response · Authors · 2023-11-21
>
> (Continued)
> > **[Q4] Can you discuss the potential development directions for evaluation frameworks that can handle the complexity of LLM-based task planning beyond simple tabletop manipulation tasks, as mentioned in the current limitation of the field?**
>
> In order to address the complexity of LLM-based task planning beyond simple tabletop manipulation tasks, it is essential to develop datasets and simulators for a variety of scenarios that focus on long-horizon tasks, similar to LoTa-Bench. The datasets we used in the present paper have 92 different home scenes in ALFRED (valid set) and 100 in WAH-NL (test set). The environment spans diverse rooms including bedrooms, kitchens, bathrooms, and living rooms. And we would like to mention that our benchmarking includes long-horizon tasks, which requires more than 25 action steps.
>
> > **Minor one: Missing references**
>
> Thank you for your pointing out these references. The first paper introduced a method for encoding an environment in the form of an object table to provide context to a language model-based planner. The second paper enhances LLM-based task planner with additional modules, eliminating irrelevant objects and receptacles from observation and tracking the completion of each step. Both works contribute to the field of LLM-based task planning by focusing on integrating contextual information to improve planning performance. We plan to add both papers to the Related Work section of our manuscript as follows:
> “Moreover, integrating context into LLM-based task planners has been shown to enhance planning efficacy (Huang et al., 2023; Yao et al., 2023; Chen et al., 2023; Lin et al., 2023; Wu et al., 2023)”
>
> ---
>
> We thank you again for your valuable feedback, and we believe that addressing these points will contribute to clarifying the contribution of the paper and guiding the direction of our research expansion. If you have any further suggestions or concerns, feel free to communicate them.

---

> ### Author Response · Authors · 2023-11-23
> **Looking Forward to Your Feedback**
>
> Dear Reviewer JeGM,
>
> Thank you again for your considerable comments for our paper. We hope that our rebuttal has addressed your concerns. As the ICLR discussion deadline approaches, we kindly request your feedback on our submitted response. If you have any further questions or require additional clarification, please let us know.
>
> Thank you very much for your time!
>
> Best wishes,
>
> The Authors

---

> > ### Comment · Reviewer_JeGM · 2023-12-04
> >
> > Thank you for the replies. I've reviewed both the other critiques and the authors' response. After considering the paper's overall quality, I prefer to maintain my current score.

---

### Official Review · Reviewer_JjuX · 2023-11-09

**Soundness:** 3 good
**Presentation:** 3 good
**Contribution:** 1 poor
**Rating:** 6
**Confidence:** 3

**Summary:**

The paper proposes LoTa-Bench, a benchmark system to automatically evaluate the performance of task planning for home-service embodied agents using large language models (LLMs) across a variety of datasets and simulators. The authors conduct extensive experiments with a baseline planner and several of its extensions within this benchmark framework.

**Strengths:**

- The paper is well-written and structured.
- The authors test multiple pre-trained LLMs (GPT, GPT-Neo series, LLaMA, OPT,MPT) on their benchmark.
- The authors present results on not just a baseline planner but also other extensions (such as feedback incorporation etc) on their benchmark.

**Weaknesses:**

The primary contribution of the paper seems limited to the integration of several existing benchmarks into a single platform. There are existing works, as cited in the Related Work section, that already employ benchmarks like ALFRED and VirtualHome to test various planning and execution techniques. The paper does not make it clear how LoTa-Bench provides additional value over these existing benchmarks. It appears that the significant novel contribution is the extension of the WAH dataset and its adaptation for autonomous agents.

**Questions:**

- Why was there no fine-tuning conducted with the WAH-NL dataset?

---

> ### Author Response · Authors · 2023-11-19
>
> Thank you for your valuable feedback. We would like to respond to your comments on the weaknesses and questions.
>
> > **The primary contribution of the paper seems limited to the integration of several existing benchmarks into a single platform. ... The paper does not make it clear how LoTa-Bench provides additional value over these existing benchmarks.**
>
> First, we clarify the differences between LoTa-Bench and existing benchmarks.
>
> * **1) LoTa-Bench vs. ALFRED + AI2THOR**
>
>     ALFRED [1] built on AI2THOR [2] has both high-level and low-level instructions. One example is as follows.
>
>     ```
>     Task instruction: "Move a towel to the sink"
>     Step-by-step action instruction:
>     1. Move to the far left cupboard under the far left sink
>     2. Open the cupboard and remove the towel inside, shut the cupboard
>     3. Carry the towel to the left sink above the cupboard
>     4. Place the towel in the sink
>     ```
>
>     As low-level instructions in natural language are given, the previous studies [3, 4] on ALFRED mostly focused on translating step instructions into robot actions with visual understanding, which overlooks the importance of high-level planning. Thus, we proposed LoTa-Bench designed to focus on high-level task planning by using only task instructions; no step-by-step instructions were given and no training on the dataset unlike previous studies to validate generalizability of the high-level task planner. Although we did not augment the ALFRED dataset itself, we utilized the dataset in a new way to propose a new benchmark problem to help the research community validate new task planning methods more easily and objectively.
>
> * **2) LoTa-Bench vs. ActivityPrograms \& Watch-and-Help + VirtualHome**
>
>     The ActivityPrograms dataset [5] provides natural language instructions, but it lacks goal conditions. Some language-oriented task planning studies that used this dataset relied on human evaluation for quantitative performance evaluation [6, 7]. LoTa-Bench, however, includes datasets with goal conditions, enabling automated quantitative performance measurement.
>
>     The Watch-and-Help dataset [8] incorporates goal conditions but not NL instructions, limiting its use for language-oriented task planning. LoTa-Bench adapts the Watch-and-Help dataset for language-oriented task planning by modifying the goal conditions for autonomous agents and enriching it with NL instructions that we collected from crowdworkers.
>
> Beyond providing benchmark suites, our comprehensive experimental analysis illuminates factors influencing LLM-based task planner performance, such as model size, type, and the number of in-context examples. We also investigate performance-enhancing extensions for LLM-based task planners within LoTa-Bench, including in-context example selection, feedback and replanning, and fine-tuning.
>
> Together, we believe that these contributions uniquely position LoTa-Bench as a valuable asset in language-oriented task planning research.
>
> [1] Alfred: A benchmark for interpreting grounded instructions for everyday tasks \
> [2] Ai2-thor: An interactive 3d environment for visual ai \
> [3] Episodic Transformer for Vision-and-Language Navigation \
> [4] Agent with the big picture: Perceiving surroundings for interactive instruction following \
> [5] Virtualhome: Simulating household activities via programs. \
> [6] Language Models as Zero-Shot Planners: Extracting Actionable Knowledge for Embodied Agents \
> [7] Parsel : Algorithmic Reasoning with Language Models by Composing Decompositions \
> [8] Watch-and-help: A challenge for social perception and human-ai collaboration

---

> ### Author Response · Authors · 2023-11-19
>
> (countinued)
>
> > **Why was there no fine-tuning conducted with the WAH-NL dataset?**
>
> In our initial approach, the primary goal of fine-tuning was to demonstrate the capability of our benchmark in facilitating a broad range of experiments for LLM-based planning. Consequently, we chose to present fine-tuning results solely within the ALFRED domain, considering it to be sufficiently illustrative of our benchmark's efficacy. Additionally, the WAH-NL domain offered a limited dataset of only 250 instances for fine-tuning, which we initially deemed inadequate for achieving significant performance improvements.
>
> Nevertheless, inspired by your query, we have now undertaken a fine-tuning experiment using the WAH-NL dataset. This dataset comprises 250 training instances, and we maintained consistency in hyper-parameters and conditions as per our ALFRED dataset experiments. The outcomes are detailed in the table below.
>
> **Table 1. Plan execution success rates of planners fine-tuned on the WAH-NL dataset. δ denotes changes in success rates with and without fine-tuning (expressed in percentage points, pp).**
> |             | Baseline Success(%) | Fine-tuned Success (%) | δ (pp) |
> |-------------|---------------------|------------------------|-------------|
> | LLaMA 1 7B  | 28.65               | 21.36                  | -7.34       |
> | LLaMA 1 13B | 23.70               | 29.47                  | +5.77       |
> | LLaMA 1 30B | 32.22               | 33.78                  | +1.56       |
>
> Although the improvements were modest and not consistent across all models, fine-tuning with the LLaMA 1 13B and 30B models did exhibit some performance enhancements (5.77 and 1.56 percent points).
>
> The entire process, excluding the time spent on fine-tuning the models, took less than 7 hours, and the evaluation was fully automated using the proposed LoTa-Bench. We are confident that our benchmark suite can significantly accelerate research on various aspects of LLM-based planners, offering an efficient and versatile evaluation tool for the research community.
>
> ---
>
> We thank you again for your valuable feedback, and we believe that addressing these points will contribute to clarifying the contribution of the paper. If you have any further suggestions or concerns, feel free to communicate them.

---

> > ### Comment · Reviewer_JjuX · 2023-11-23
> >
> > Thanks a lot for your response! Most of my concerns have been addressed and I have updated the score of the paper. I hope that the authors emphasize on their contributions more clearly in the paper (for example, by juxtaposing the existing benchmarks and the additions/integrations that have been done on top of them in the introduction section itself).

---

> ### Author Response · Authors · 2023-11-23
> **Looking Forward to Your Feedback**
>
> Dear Reviewer JjuX,
>
> Thank you again for your considerable comments for our paper. We hope that our rebuttal has addressed your concerns. As the ICLR discussion deadline approaches, we kindly request your feedback on our submitted response. If you have any further questions or require additional clarification, please let us know.
>
> Thank you very much for your time!
>
> Best wishes,
>
> The Authors

---

### Author Response · Authors · 2023-11-22
**Overall Response to All Reviewers (Revision Uploaded)**

We appreciate all reviewers for your valuable time and insightful comments. In response to your feedback, we have carefully revised our paper and highlighted the changes in blue in our revised draft. Here is a summary of the main revisions:

* Fine-tuning on WAH-NL train set (**Section 6.3 & Appendix G**): In line with Reviewer JjuX's insightful suggestions, we conducted additional experiments related to fine-tuning on the WAH-NL train set. The results are thoroughly discussed in Section 6.3, with detailed findings presented in Appendix G.

    [Reviewer JjuX's Comments](https://openreview.net/forum?id=ADSxCpCu9s&noteId=3zoH0Xtbjf)

* Enhancing literature references (**Section 2**): Following Reviewer JeGM’s recommendations, we have enriched Section 2 with references to studies that explore the integration of context information into LLM-based task planners.

    [Reviewer JeGM's Comments](https://openreview.net/forum?id=ADSxCpCu9s&noteId=ayZME4jrnc)

* In-depth analysis of ALFRED's failure cases (**Section 5.2**): Addressing Reviewer DjjY's comments, we have analyzed the failure cases of ALFRED, providing a deeper understanding of the challenges that LLM-based task planners faced.

    [Reviewer DjjY's Comments](https://openreview.net/forum?id=ADSxCpCu9s&noteId=RpOihxyNvX)

* Experiments using GPT-4 (**Section 5.2 & Appendix D**): In response to Reviewer DjjY’s valuable advice, we included experiments utilizing GPT-4. These results are summarized in Section 5.2, with comprehensive details in Appendix D.

    [Reviewer DjjY's Comments](https://openreview.net/forum?id=ADSxCpCu9s&noteId=xV88n6McBv)

* Comprehensive experiments with full set of ALFRED (**Section 5.1 & Appendix C**): Heeding Reviewer mB86's concern, we extended our main experiments using the full set of ALFRED instead of the previously used 30% subset. This is detailed in Appendix C and mentioned in Section 5.1.

    [Reviewer mB86's Comments](https://openreview.net/forum?id=ADSxCpCu9s&noteId=ZFRpQZrFZh)

* Font corrections: We addressed the different font issue highlighted by Reviewer mB86 by modifying our compiling options to align with the original ICLR format, ensuring a uniform presentation in the revised draft.

    [Reviewer mB86's Comments](https://openreview.net/forum?id=ADSxCpCu9s&noteId=qyPOhhyuPf)

We hope that these revisions adequately address your concerns and contribute to the improvement of our manuscript. We look forward to any further suggestions or comments.

---

### Meta-Review · Area_Chair_XAg7 · 2023-12-17

**Metareview:**

LLMs have emerged as a common baseline for embodied AI agents, specifically for improving task planning. Each benchmark has its own action space, set of tasks, and so forth making it difficult for a single policy to be tested across a suite of benchmarks -- this work aims to address this and in turns allows for a new set of model evaluations to understand what decisions might be broadly applicable/useful vs domain specific.

**Justification For Why Not Higher Score:**

The work targets a somewhat constrained setting, for example: two benchmarks, only high level task planning, and not the full versions of either dataset.

**Justification For Why Not Lower Score:**

The construction of infrastructure on top of two simulators to provide for a head-to-head model comparison is a substantial engineering lift and allowed for comparisons not previously possible but clearly useful (if even just to confirm intuitions)

---

### Decision · Program_Chairs · 2024-01-16

Accept (poster)